# Dissolved Mn(III) is a key redox intermediate in sediments of a seasonally euxinic coastal basin

Robin Klomp[1,2], Olga M. Żygadłowska[2], Mike S.M. Jetten[1], Véronique E. Oldham[3], Niels A.G.M. van Helmond[1,2], Caroline P. Slomp[1,2] and Wytze K. Lenstra[1,2]

[1]Department of Microbiology, Radboud Institute for Biological and Environmental Science, Radboud University, Heyendaalseweg 135, 6525 AJ Nijmegen, The Netherlands
[2]Department of Earth Sciences, Utrecht University, Princetonlaan 8a, 3584 CB Utrecht, The Netherlands
[3]Graduate School of Oceanography, Rhode Island University, 215 S Ferry Rd, Narragansett, RI 02882, U.S.A.

*Correspondence to*: Robin Klomp (robin.klomp@ru.nl)

**Abstract.** Manganese (Mn) is an essential micronutrient and key redox intermediate in marine systems. The role of organically complexed dissolved Mn(III) (dMn(III)-L) as an electron acceptor and donor in marine environments is still incompletely understood. Here, we use geochemical depth profiles of solutes and solids for the sediment and overlying waters and a reactive transport model to reconstruct the seasonality in sedimentary dMn(III)-L dynamics and benthic Mn release in a eutrophic, seasonally euxinic coastal basin (Lake Grevelingen, the Netherlands). Our model results suggest that dMn(III)-L is a major component of the dissolved Mn pool throughout the year. According to the model, there are three major sources of pore water dMn(III)-L when oxygen ($O_2$) is present in the bottom water, namely reduction of Mn oxides coupled to the oxidation of Fe(II), reduction of Mn oxides coupled to organic matter degradation and oxidation of Mn(II) with $O_2$. Removal of pore water dMn(III)-L is inferred to primarily take place through reduction by dissolved Fe(II). When bottom waters are euxinic in summer, model-calculated rates of sedimentary Mn cycling decrease strongly, because of a lower supply of Mn oxides. The dMn(III)-L transformations in summer mostly involve reactions with Fe(II) and organic matter. Modelled benthic release of Mn mainly occurs as dMn(III)-L when bottom waters are oxic, as Mn(II) upon initial bottom water euxinia and as both Mn(II) and dMn(III)-L when the euxinia becomes persistent. Our model findings highlight strong interactions between the sedimentary Fe and Mn cycles. Dissolved Mn(III)-L is a relatively stable and mobile Mn species, when compared to Mn(II), and is therefore more easily transported laterally throughout the coastal zone and possibly also to open marine waters.

## Keywords

Manganese dynamics, benthic flux, reactive transport modelling, micronutrient

## 1 Introduction

Manganese (Mn) is an essential micronutrient and is one of the most abundant transition metals in natural environments (Raven, 1990; Neretin et al., 2003). In marine systems, dissolved Mn is present as either Mn(II) or Mn(III) complexed to organic ligands (dMn(III)-L) (Burdige, 1993; Luther, 2010; Madison et al., 2013). Importantly, dMn(III)-L can act as either an electron acceptor or donor and can thereby play a key role as a redox intermediate in marine sediments (Kostka et al., 1995; Trouwborst et al., 2006; Madison et al., 2013). Sediments may act as a source of both dissolved Mn(II) and dMn(III)-L to overlying waters (Oldham et al., 2019). The redox state of the dissolved Mn will determine its reactivity and mobility and thus its ultimate fate in the water column (Oldham et al., 2017a; Lenstra et al., 2020).

Sedimentary Mn cycling is driven by interactions between Mn and other redox sensitive compounds (Burdige, 1993). For example, reductive dissolution of Mn oxides, which leads to mobilization of dissolved Mn, can be coupled to the oxidation of organic matter (OM), hydrogen sulfide ($H_2S$), ferrous iron (Fe(II)) or methane ($CH_4$) (Postma, 1985; Burdige, 1993; Aller, 1994; Beal et al., 2009). Interactions between Mn oxide and ammonium ($NH_4^+$) have also been proposed (Hulth et al., 1999; Thamdrup and Dalsgaard, 2000). The occurrence of this process in marine environments is still debated, however. Dissolved Mn may be in the form of Mn(II) or dMn(III)-L (Madison et al., 2013; Luther et al., 2018). Dissolved Mn(II) can precipitate as Mn-carbonate when alkalinity is high (Calvert and Pedersen, 1996; Lepland and Stevens, 1998) and can adsorb onto Mn oxide minerals (van der Zee et al., 2001).

In the presence of oxygen ($O_2$), for example in surface sediments in marine settings with oxygenated waters, dissolved Mn(II) and dMn(III)-L can be oxidized, forming Mn oxides. When there is little $O_2$ penetration into the sediment or when macrofauna are present, the dissolved Mn may escape to the overlying water (Slomp et al., 1997; McManus et al., 2012; Lenstra et al., 2020). In oxic waters, dissolved Mn may oxidize to Mn oxides and settle under gravitational forcing thereby enhancing the supply of Mn oxides to the sediment (Sulu-Gambari et al., 2017; Lenstra et al., 2021a). When bottom waters are anoxic, sediments will eventually become depleted of reactive Mn oxides and dissolved Mn can accumulate in the water column  (Lenz et al., 2015; Dellwig et al., 2018).

Part of the Mn released from the sediment could consist of dMn(III)-L (Oldham et al., 2019). Dissolved Mn(III) is highly reactive, but the complexation with organic ligands can result in a meta-stable complex (Kostka et al., 1995). The strength of the bond between Mn(III) and the ligand determines the reactivity of dMn(III)-L (Luther et al., 2015; Oldham et al., 2015, 2017b). The Mn(III)-L complex can be the dominant form of dissolved Mn in sediment pore water and in the water column (e.g. Trouwborst et al., 2006; Madison et al., 2013; Oldham et al., 2017b). Oxidation of Mn(II) and reduction of Mn(IV) are suggested to occur via one-electron step transitions with dMn(III)-L as an intermediate product (Luther, 2005). Formation of dMn(III)-L has been proposed to mainly take place via reduction of Mn oxides coupled to OM degradation or oxidation of Mn(II) by $O_2$ (Madison et al., 2013). Other pathways that produce dMn(III)-L include reduction of Mn oxides coupled to oxidation of Fe(II) and $H_2S$ (Madison et al., 2013). Removal of dMn(III)-L is suggested to mainly occur via oxidation by $O_2$ to Mn(IV) (Madison et al., 2013). Reduction of dMn(III)-L by oxidation of Fe(II), $H_2S$, nitrite ($NO_2^-$) and OM is also

possible (Kostka et al., 1995; Oldham et al., 2015, 2019; Karolewski et al., 2021), but does not necessarily always occur (Oldham et al., 2015, 2019).

In recent years, the number of areas that experience eutrophication and deoxygenation has increased strongly (Diaz & Rosenberg, 2008; Breitburg et al., 2018). Anoxia initially stimulates benthic release of dissolved Mn, since the dissolved Mn will no longer be re-oxidized near the sediment-water interface (Pakhomova et al., 2007; Lenstra et al., 2021a). Furthermore,

organic carbon ($C_{org}$) oxidation rates and the benthic release of dissolved Mn can be positively correlated (Berelson et al., 2003; McManus et al., 2012). At present we do not know to what extent dMn(III)-L contributes to dissolved Mn released from the sediment and what processes control the redox state in which Mn leaves the sediment.

The aim of this study is to understand the effects of seasonal euxinia on dMn(III)-L dynamics. This is studied in sediments in a seasonally euxinic coastal basin (Lake Grevelingen, the Netherlands). We combine detailed sediment and pore water

analyses with a reactive transport model that, for the first time, includes a detailed Mn(III) cycle, to investigate the main drivers of sedimentary Mn cycling and the benthic release of Mn. Our model results suggest that dMn(III)-L is released from the sediment both when bottom waters are oxic and euxinic. Furthermore, our results highlight the importance of dissolved Fe(II) for the mobilization of Mn and release of dissolved Mn from the sediment in seasonally euxinic basins.

**2 Methods**

**2.1 Study area**

Lake Grevelingen is a coastal marine system in the south-west of the Netherlands (Fig. 1). It is a former estuary that was closed by a dam on the landward side in 1964 and a dam on the seaward side in 1971, in response to major flooding in the area. A sluice on the seaward side enables sea water exchange with the North Sea. The lake has an average depth of 5.1 m but is intersected by former tidal channels with several deeper basins (Egger et al., 2016).

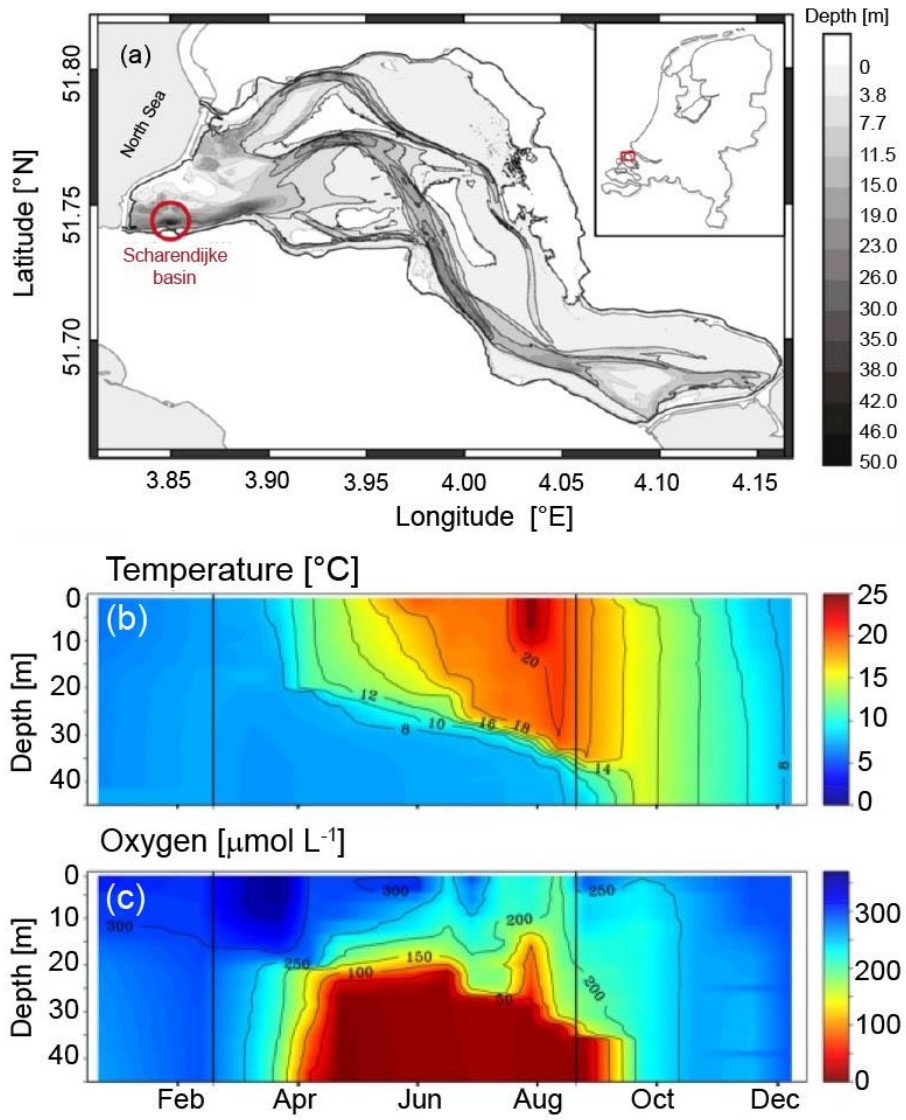

**Figure 1. (a) Map of Lake Grevelingen indicating the location of the Scharendijke basin (adapted from Egger et al. (2016)). (b) Temperature and (c) O₂ concentrations as measured by Rijkswaterstaat (Directorate-General for Public Works and Water Management of the Netherlands). The observations by Rijkswaterstaat were carried out every two weeks to one month, giving a total of 19 time points in 2020 in the water column of the Scharendijke basin throughout 2020 (Adapted from Żygadłowska et al. 2023)). The black vertical lines indicate the dates of sampling in March and September.**

In this study, we focus on Scharendijke basin, located in the deepest part of the lake (Fig. 1; 51.742°N, 3.849°E; water depth of 45 m). During summer, temperature-driven stratification of the water column leads to the development of anoxic and sulfidic bottom waters, as recorded by the seasonal drawdown of molybdenum from the water column and its consequent sedimentary enrichment (Egger et al., 2016; Żygadłowska et al., 2023, 2024b). Water column euxinia was confirmed by direct measurements of $H_2S$ in 2021 (Żygadłowska et al., 2024a, b). Each year, the water column mixes again in autumn, resulting in bottom water reoxygenation. Scharendijke basin is a relatively narrow and deep basin in an overall shallow lake (Figure 1). As a consequence, vertical transport settling of suspended matter is expected to be supplemented by lateral transport of material from shallower areas near the sediment-water interface. The sediment at this site is characterized by a high sedimentation rate (~13 cm yr$^{-1}$) and high rates of OM oxidation (~249 mmol C m$^{-2}$ d$^{-1}$) which is also reflected in a shallow sulfate-methane transition zone at around 20 cm depth in the sediment (Egger et al., 2016). High sulfate reduction rates lead to significant build-up of $H_2S$ in the sediment in summer (Egger et al., 2016). Macrofauna are absent from the sediment in Scharendijke basin (based on visual observations of sediment sieved over 0.5 mm). In 2020, two field campaigns were carried out with RV *Navicula*, one when the water column was mixed and bottom waters were oxygenated (March) and one when the water column was stratified and bottom waters were euxinic (September). During these field campaigns, the water column and sediment was sampled as described in detail below. Additional sampling was carried out in eight field campaigns, one in each month from March to October 2021, as described in the supplement (Section 1).

## 2.2 Water column sampling

Depth profiles of temperature and $O_2$ were obtained from the water column with a CTD (Seabird SBE 911 plus) equipped with an oxygen sensor (SBE43) during the cruise in September 2020. Most $O_2$ sensors, including the SBE43, measure a background signal when $O_2$ is absent. Winkler titrations cannot be used for calibration, because these also have artefacts when $O_2$ concentrations are low (Grégoire et al., 2021). Therefore we assume absence of $O_2$ when oxygen concentrations are low and concentrations do not change with increasing water depth (Żygadłowska et al., 2023). Water samples were collected with a 10 L Niskin bottle at discrete depths (1-5 m depth resolution) in September 2020. Samples were obtained for analysis of Mn(II) and dMn(III)-L and stored in a $N_2$ purged aluminum bag at -20°C until analysis.

## 2.3 Sediment and pore water collection

During both sampling campaigns in 2020, four sediment cores were collected with a UWITEC gravity corer with transparent PVC core liners of 120 cm length with a 6 cm inner diameter. During sampling, the surface sediment remained intact. The first core was sectioned for pore water and solid phase analyses at a depth resolution of 1 cm in a glove bag under a $N_2$ atmosphere on board the ship. The sediment was placed in 50 ml centrifuge tubes and subsequently centrifuged at 4500 rpm for 20 minutes to separate the pore water from the solid phase. Pore water was collected for all samples in the upper 10 cm,

for samples from every second cm between 10 and 50 cm and every fifth cm from 50 cm until the bottom of the core. The supernatant was filtered over 0.45 μm pore size filters (i.e. capturing the aqueous and colloidal fractions; Raiswell and Canfield (2012) under a $N_2$ atmosphere in a glove bag and subsequently sub-sampled for the analysis of $NH_4^+$, sulfate ($SO_4^{2-}$), alkalinity, $H_2S$, total dissolved Fe and Mn and dissolved Mn(II) and dMn(III)-L. The samples for $H_2S$ analysis were diluted five times in a 2% Zn-acetate solution in a glass vial and stored at 4°C. Samples for the analysis of $NH_4^+$ and, in

September, for the analysis of $NO_2^-$ and nitrate ($NO_3^-$) were stored at -20°C. The samples for $SO_4^{2-}$ and alkalinity were stored at 4°C. Samples for total dissolved Fe and Mn (TD Fe and TD Mn) were acidified with 10 μL 35% suprapure HCl per ml of sample and stored at 4°C. Samples for the analysis of dissolved Mn(II) and dMn(III)-L and the sediment residues were stored in $N_2$ purged aluminum bags at -20°C.

The second core, with pre-drilled holes at 2.5 cm intervals covered with tape prior to coring, was used directly after retrieval

to collect samples for pore water $CH_4$ concentrations. Plastic cut off syringes were used to transfer 10 ml of sediment directly into 65 ml glass bottles filled with saturated NaCl solution. The bottles were then stoppered, capped and stored upside down until analysis. Note that degassing of $CH_4$ during the sampling may lead to an underestimation of the total $CH_4$ concentrations, especially when $CH_4$ concentrations are high (Egger et al., 2017; Jørgensen et al., 2019). The third core was sectioned at a resolution of 1 cm intervals to determine sediment porosity. The sediment was placed into pre-weighed 50 ml

centrifuge tubes. The fourth core was used for high-resolution depth profiling of $O_2$ with micro-electrodes as described in (Żygadłowska et al., 2023).

**2.4 Chemical analysis of pore water and water column samples**

Pore water $NH_4^+$ was analyzed spectrophotometrically with the indophenol blue method (Solórzano, 1968). Concentrations

of $NO_2^-$ and $NO_3^-$ were measured with a Gallery™ Automated Chemistry Analyzer type 861 (Thermo Fisher Scientific; detection limit of 1 μmol $L^{-1}$). Concentrations of $SO_4^{2-}$ were measured using ion chromatography (Metrohm 930 Compact IC Flex; detection limit for $SO_4^{2-}$ of 50 μmol $L^{-1}$). Alkalinity was measured through titration with 0.01 M HCl, within 24 h of sampling. Samples for $H_2S$ were determined spectrophotometrically, using an acidified solution of phenylenediamine and ferric chloride, where $H_2S$ is the sum of $S^{2-}$, $HS^-$ and $H_2S$ (Cline, 1969). Total dissolved Fe and Mn were determined by

Inductively Coupled Plasma Optical Emission Spectroscopy (ICP-OES, Perkin Elmer Avio; detection limit 0.1 μmol $L^{-1}$ and 0.03 μmol $L^{-1}$ for Fe and Mn respectively).

Dissolved Mn(II) and dMn(III)-L concentrations were determined simultaneously via a kinetic spectrophotometric method using porphyrin, cadmium chloride and an imidazole borate buffer as described previously (Madison et al. 2011; detection limit of 1 μmol $L^{-1}$ for Mn). The kinetics of the Mn(II) – porphyrin reaction depends on environmental characteristics such

as salinity (Thibault de Chanvalon and Luther, 2019) and should therefore be determined for each site separately. Here, the kinetic constant value for Mn(II) was determined in triplicate on an aliquot of sample in which the dissolved Mn was completely reduced by adding an excess of hydroxylamine (Oldham et al., 2015; for kinetic curves of the triplicate analysis

see Fig. S1). All samples were analyzed in triplicate using a 1 cm pathlength quartz cuvette in a Shimadzu UV-1900i spectrophotometer (for examples of the kinetic curves, see Fig. S2). Our analyses were all carried out under normal atmospheric conditions. Strongly bound Mn(III)-ligand complexes cannot be measured via this method (Oldham et al., 2015; Kim et al., 2022). Therefore, the difference between the sum of measured Mn(II)/Mn(III)-L and the total dissolved Mn measured by ICP-OES can be interpreted as the amount of Mn(III)-L that is bound to strong ligands.

Prior to the analysis of $CH_4$, a 10 ml $N_2$ headspace was injected in the bottle. After seven days, when equilibrium between the water and gas phase was established, $CH_4$ was measured with a Thermo Finnigan Trace™ gas chromatograph (Flame Ionization Detector; limit of detection 0.02 µmol $L^{-1}$).

## 2.5 Solid phase analyses

Porosity was determined based on the weight loss upon drying the samples in an oven at 60°C. Sediment residues for the anoxic analyses were freeze-dried and subsequently ground with an agate mortar and pestle under a $N_2$ environment. For analysis of $C_{org}$ content, aliquots of ca. 300 mg of the powdered sediment were decalcified with 1 M HCl (2-step wash; Van Santvoort et al., 2002), dried, weighed and powdered. The aliquot was analyzed with an elemental analyzer (Fisons Instruments model NA 1500 NCS) and the C content was corrected for the weight loss during decalcification. The accuracy and precision of the analyses was determined based on measurements of the internationally certified soil standard IVA2. The certified value for IVA2 is 0.732 wt.% C. The mean value that was obtained in this study for IVA2 (n=24) was 0.722 wt.% C, with a standard deviation of 0.009 wt.% C. The analytical uncertainty based on duplicates (n=15) was 0.11 wt.% C for organic C.

A second aliquot (50 – 100 mg) was analyzed via a sequential extraction procedure to determine the speciation of Fe following a combination of the methods from Poulton and Canfield (2005) and Claff et al. (2010) as described by Kraal et al. (2017) and Mn (Lenstra et al., 2021b). The extraction procedure consists of the following five steps: (1) a mixture of 0.057 M ascorbic acid, 0.17 M sodium citrate and 0.6 M sodium bicarbonate, pH of 7.5, to extract Fe oxides and easily reducible Mn, (2) 1 M HCl to extract reducible crystalline Fe oxides, Fe carbonate, FeS and Mn carbonate, (3) 0.35 M acetic acid, 0.2 M $Na_3$ citrate and 50 g $L^{-1}$ Na dithionite, pH 4.8, to extract crystalline Fe and Mn oxides, (4) 0.2 M ammonium oxalate and 0.17 M oxalic acid to extract recalcitrant Fe oxides and non-reactive Mn such as Mn associated with clay minerals, (5) 65% $HNO_3$ was used to extract pyrite and Mn associated with pyrite. Extracted Fe and Mn in step 1, 2 and 5 was measured with ICP-OES (ICP-OES, Perkin Elmer Avio; detection limit 0.1 µmol $L^{-1}$ and 0.03 µmol $L^{-1}$ for Fe and Mn respectively). The average recovery for Fe and Mn was 106% and 100%, respectively and the average analytical uncertainty based on duplicates (n=16) was 3.2% and 2.6%, respectively. Extracted Fe in steps 2, 3 and 4 was determined spectrophotometrically using the phenanthroline method (APHA, 2005). Average analytical uncertainty based on duplicates (n=16) was < 13.4% for all fractions of the three sequential extraction procedures. Mn extracted in step 3 and 4 is not measured, because this is

190 mainly Mn associated with clays (Lenstra et al., 2021), which is not relevant in this study. The concentration of Fe oxides is assumed to be the sum of the Fe extracted in steps 1, 3 and 4. The Fe extracted in step 2 is not taken into account for Fe oxides, because a separation of Fe(II) and Fe(III) on selected samples, to separate Fe oxides from Fe(II) containing minerals like FeS and Fe carbonate, indicated that nearly all Fe in this step was present as Fe(II). The concentration of Mn oxides is assumed to be the Mn extracted in step 1 (Anschutz et al., 2005; Lenstra et al., 2021b).

## 2.6 Calculation of benthic diffusive fluxes

Diffusive fluxes of dissolved Mn across the sediment-water interface were calculated with Fick's law of diffusion, based on the gradient in total Mn concentration between the bottom water and the pore water in the upper cm of the sediment (at an average depth of 0.5 cm) by applying the formula:

$$J = -\varphi D_s \frac{dC}{dz} \qquad (1)$$

where J is the diffusive flux in mmol $m^{-2}$ $d^{-1}$, $\phi$ is the porosity of the sediment, Ds is the diffusion coefficient for Mn in the sediment in $m^{-2}$ $d^{-1}$, C is the concentration of Mn in $\mu$mol $m^{-3}$ and z is the sediment depth in m. In our calculations, we assumed all Mn was present as Mn(II), The Ds for Mn(II) was corrected for temperature and salinity using the R package CRAN: marelac (Soetaert et al., 2010), taking into account the tortuosity of the sediment (Boudreau, 1996).

## 2.7 Reactive transport model

A 1-dimensional reactive transport model, written in R (version 3.6.2) and modified from (Egger et al., 2016; Lenstra et al., 2018), was used to model the sedimentary Mn cycle, including the dynamics of both dissolved Mn(II) and dMn(III)-L. The model, a standard multicomponent reactive transport model, is based on the principles that are outlined in, for example, Van Cappellen and Wang (1996). The modelled components include 9 solids and 12 solutes (Table S1). Solids are transported by advection (burial), while solutes are transported both by advection and molecular diffusion. The model includes 36 reactions (Table S2, S3). The reaction parameters were obtained from the literature or constrained using the model (Table S3, S4). Chemical and physical constants were calculated using the *marelac* package (Soetaert et al., 2010) and transport coefficients were calculated using the *reactran* package (Soetaert and Meysman, 2012) and were, where relevant, adjusted for environmental characteristics at the study site (Table S5). The diffusion coefficient for dMn(III)-L was set to a lower value than that of Mn(II), because the diffusive behavior of metal-ligand complexes is typically controlled by the organic ligand (Furukawa and Takahashi, 2008; Table S6).

The model describes the upper 100 cm of the sediment column, which is divided into 1000 model layers of 1 mm. The boundary conditions are set at the top of the sediment and are defined as fixed concentrations for solutes and fluxes for solids (Table S7). The model was fit to the data set for 2020. In a spin up of the model, the boundary conditions were fixed until

steady state was reached after 60 years, as in Egger et al. (2016). The model was then run for 20 years, in which the seasonal cycle of oxic – euxinic conditions was simulated by varying the bottom water $O_2$ and $H_2S$ concentration, the influx of Fe-oxides, Mn oxides and OM and the sedimentation rate (Fig. S3). The key purpose of the model application was to determine the seasonality in the production and removal pathways of dMn(III)-L in the sediment and estimate the rate of diffusive release of dMn(III)-L and Mn(II) to the overlying water. Here we specifically focus on the general trends in the key

processes that regulate the seasonal dMn(III)-L dynamics in a eutrophic basin. As the model focusses on these general trends, the overall picture of Mn dynamics at this site will not change due to uncertainties related to, for example, the sample resolution in the top part of the sediment.

In the model, high rates of $CH_4$ production lead to an overestimation of $CH_4$ concentrations because $CH_4$ bubble formation is not included. Adding bubble formation would not improve our main model results regarding dMn(III)-L dynamics, however,

but would increase model uncertainty. We note that a perfect model fit of the data is not expected, due to the complexity of the system and the strong seasonal variations at our study site which also vary between years. This is especially true for the fit to the solid phase profiles since such variations in the reactivity and input flux of solid phases are not specifically included in the model (Table S4, S5). A detailed model description is provided in supplement section 2.

A sensitivity analysis was performed to investigate the response of the benthic release of dMn(III)-L and Mn(II) to variations

in OM input. The input of OM was varied by a factor of 0.01 to 2, relative to the base line scenario, leading to average OM oxidation rates ranging from 0.6 to 143 mmol C $m^{-2}$ $d^{-1}$ in the months when bottom waters were oxygenated and 0.9 to 219 mmol C $m^{-2}$ $d^{-1}$ in months with euxinic conditions. To obtain insight in the processes controlling the transformation and benthic release of Mn, reaction rates were integrated over the upper 10 cm of the sediment.

A forward simulation of pore water and solid phase data collected during 8 sampling events, one in each month from March

to October 2021, was performed to verify the model.

# 3 Results

## 3.1 Water column and pore water profiles

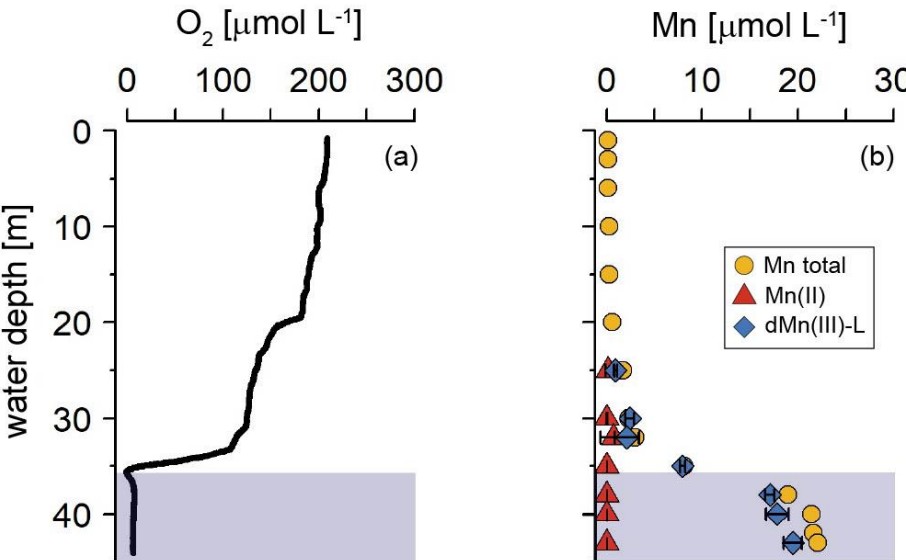

**Figure 2. A) Water column O₂ and B) total dissolved Mn and dissolved Mn(II) and dMn(III)-L (error bars represent standard deviation; n=3) in the Scharendijke basin as observed in September 2020. The shaded area indicates the anoxic part of the water column.**

In March 2020, the water column of Scharendijke basin was fully oxygenated (Fig. 1). In September, in contrast, $O_2$ depletion was observed below a water depth of 35 m (Fig. 2) and dissolved Mn, primarily present as dMn(III)-L, accumulated to concentrations of up to 22 μmol $L^{-1}$ in the anoxic waters (Fig. 2). Small enrichments in Mn(II) and dMn(III)-L were observed above the redox cline at depths of 32 and 30 m, respectively.

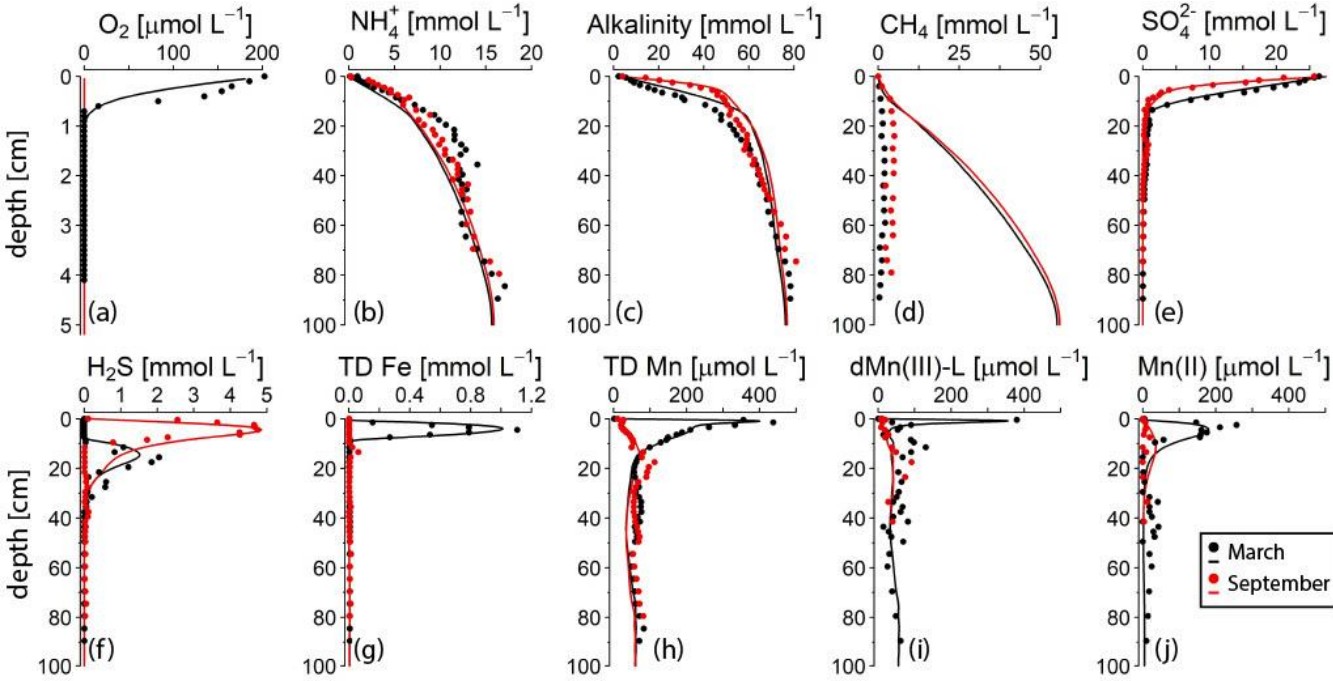

**Figure 3. Pore water profiles of key components in March (black) and September (red) 2020. The dots represent measured concentrations, the lines indicate the results of the reactive transport model. Note the different depth scale on the y-axis for $O_2$. TD Fe and TD Mn refer to total dissolved Fe and total dissolved Mn. Profiles of Mn(II) and dMn(III)-L with standard deviation error bars (n =3) and the contribution to TD Mn (in %) can be found in Fig. S5, S6. No $O_2$ was detected in the sediment in September 2020. A zoom of the top 20 cm of the profiles is presented in Fig. S7.**

The seasonal contrast in bottom water oxygen was reflected in the $O_2$ concentration in the sediment: while $O_2$ was present in the upper 0.6 cm of the sediment in March 2020, $O_2$ was absent in September 2020 (Fig. 3; for a zoom of the top 20 cm, see Fig. S7). In September, $NO_2^-$ and $NO_3^-$ did not exceed the detection limit of 1 µmol $L^{-1}$ and showed no trend with depth. Concentrations of $NH_4^+$, alkalinity and $CH_4$ increased with sediment depth to maximum values of ~15, 80 and 4 mmol $L^{-1}$, respectively. In the surface sediment (upper 10 cm) the profiles of $NH_4^+$ and alkalinity showed a distinct seasonality, however, with higher concentrations in September compared to March. Profiles of $SO_4^{2-}$ and $H_2S$ also varied between the two seasons seasonally with $SO_4^{2-}$ being removed at a shallower depth in September than in March and the zone of high $H_2S$ concentrations (i.e. 1.5 to 5 mmol $L^{-1}$) shifting upwards by 10 cm. Dissolved Fe, in contrast, was abundantly present in the top 10 cm of the sediment in March, even reaching values of up to 1.1 mmol $L^{-1}$, but was nearly absent from the pore water in September. Similarly, concentrations of dissolved Mn reached a maximum of 437 µmol $L^{-1}$ near the sediment-water interface in March but were much lower in September. The peak in dissolved Mn in the upper 10 cm of the sediment in March was found to consist of two partially overlapping peaks, with that of dMn(III)-L (up to 380 µmol $L^{-1}$) explaining the

sharp rise in dissolved Mn in the top cm of the sediment, and a broad peak in dissolved Mn(II) (up to 257 $\mu$mol L$^{-1}$) accounting for most (generally >75%; Fig. S5) of the remaining Mn in the top 10 cm.

Between 10 and 30 cm, dissolved Mn was mainly present as dMn(III)-L (up to 100% of the total dMn pool) and below 30 cm dissolved Mn(II) and dMn(III)-L contributed equally, both varying between 25% and 75% of the total dMn pool. In September, dissolved Mn was mainly present in the form of dMn(III)-L, which almost always accounted for >50% of the total dMn pool (Fig. S6).

The diffusive benthic Mn fluxes calculated with Fick's law of diffusion, based on the concentrations of the total dissolved Mn, were 2.1 mmol m$^{-2}$ d$^{-1}$ in March and 0.09 mmol m$^{-2}$ d$^{-1}$ in September.

## 3.2 Solid phase profiles

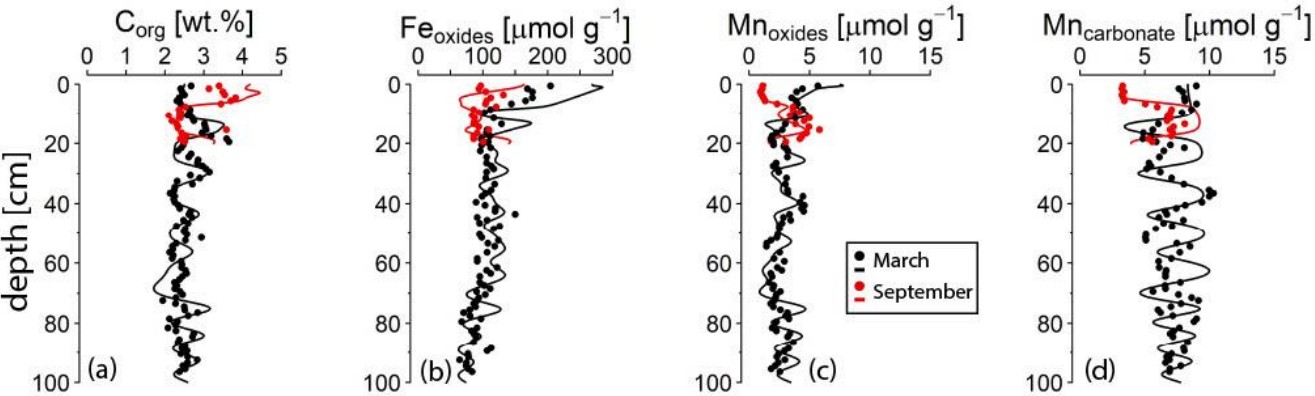

**Figure 4. Solid phase profiles of key components in March (black) and September (red) 2020. The dots represent measured values, the lines indicate the results of the reactive transport model. Fe oxides refer to the sum of the Fe extracted in the ascorbic acid, CDB and ammonium oxalate extraction steps. Mn oxides refer to the Mn that is extracted in the ascorbic acid step. The profiles for all extraction steps are shown in Fig. S8 and S9.**

The solid phase profiles of C$_{org}$, Fe- and Mn oxides and Mn carbonate reflect the strong seasonality of biogeochemical processes in the basin. The C$_{org}$ content in the top 10 cm of the sediment was much lower in March 2020 (~2.5 wt%) compared to September 2020 (~3.5 wt%) (Fig. 4). In March, when O$_2$ was present in the bottom water, the top part of the sediment was enriched in Fe and Mn oxides and Mn carbonate. In September, when bottom waters were euxinic, surface enrichments in Fe and Mn oxides and Mn carbonate were absent (Fig. 4). The oscillations in the solid phase records are preserved upon burial of the sediment. For profiles of all the Fe and Mn fractions, we refer to the supplement (Fig. S8, S9). Porosity values vary between 0.98 and 0.88, with a general trend towards lower values deeper in the sediment (Fig. S8, S9).

## 3.3 Sedimentary reactive transport modelling

### 3.3.1 Model fit

The reactive transport model generally describes the depth trends in the pore water and solid phase profiles for March and September 2020 quite well (Fig. 3, 4). For the pore water, the only exceptions are the modelled $CH_4$ profile and the profile of dMn(III)-L between 10 and 20 cm. For the solid phase profiles, the amplitude of the change in $C_{org}$, Fe oxide and Mn carbonate content is not always fully captured. In the next sections, we will primarily focus on a description of the model results for 2020 that are relevant to dMn(III)-L dynamics.

### 3.3.2 Dissolved Mn(III)-L dynamics in the sediment

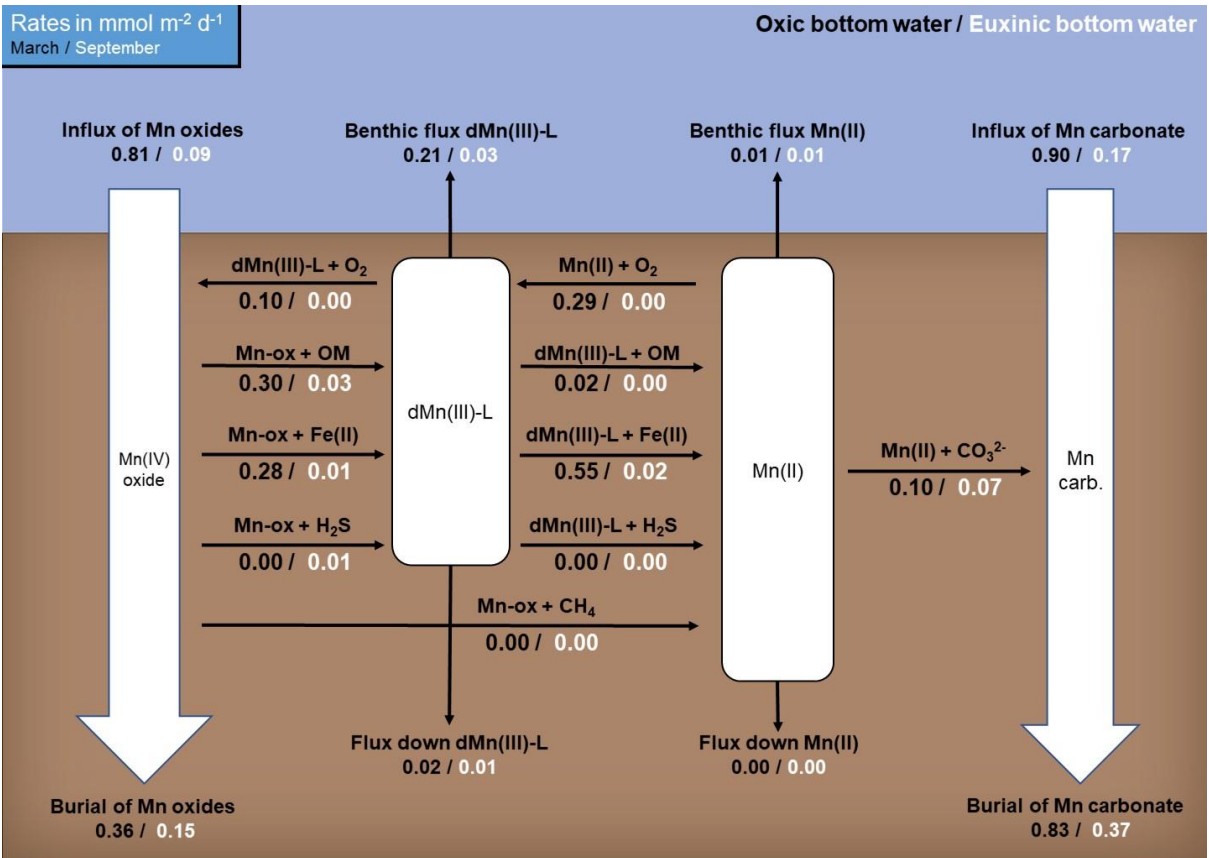

**Figure 5. Rates of Mn cycling in March (black) and September (white) 2020 as calculated with the reactive transport model. Numbers represent depth integrated reaction rates in mmol m$^{-2}$ d$^{-1}$. Note that the rates are not balanced because the system is not at steady state. The depth profiles of the reaction rates can be found in Fig. S10.**

Depth-integrated reaction rates show that, according to the model, the formation of dMn(III)-L in March 2020 is driven equally by reduction of Mn oxides coupled to OM degradation and Fe(II) oxidation and oxidation of dissolved Mn(II) by $O_2$ (each ~0.3 mmol m$^{-2}$ d$^{-1}$; Fig. 5; for reaction rate profiles see Fig. S10) with a negligible role for reduction of Mn oxides by H$_2$S. Removal of dMn(III)-L in March occurs via reduction coupled to Fe(II) oxidation (0.55 mmol m$^{-2}$ d$^{-1}$), oxidation by O$_2$ (0.1 mmol m$^{-2}$ d$^{-1}$) and benthic release (0.21 mmol m$^{-2}$ d$^{-1}$). Besides oxidation by O$_2$, dissolved Mn(II) precipitates as Mn carbonate and is released to the water column. In September 2020, the input of Mn oxides is 9 times lower than in March, leading to much lower rates of Mn cycling. Formation of dMn(III)-L is still coupled to OM degradation (0.03 mmol m$^{-2}$ d$^{-1}$) and oxidation of Fe(II) (0.01 mmol m$^{-2}$ d$^{-1}$) but now oxidation of H$_2$S also contributes (0.01 mmol m$^{-2}$ d$^{-1}$). Removal of dMn(III)-L in September mainly takes place via benthic release (0.03 mmol m$^{-2}$ d$^{-1}$) and reduction of dMn(III)-L coupled to oxidation of Fe(II) (0.02 mmol m$^{-2}$ d$^{-1}$). Again, Mn(II) is removed as Mn carbonate and via benthic release.

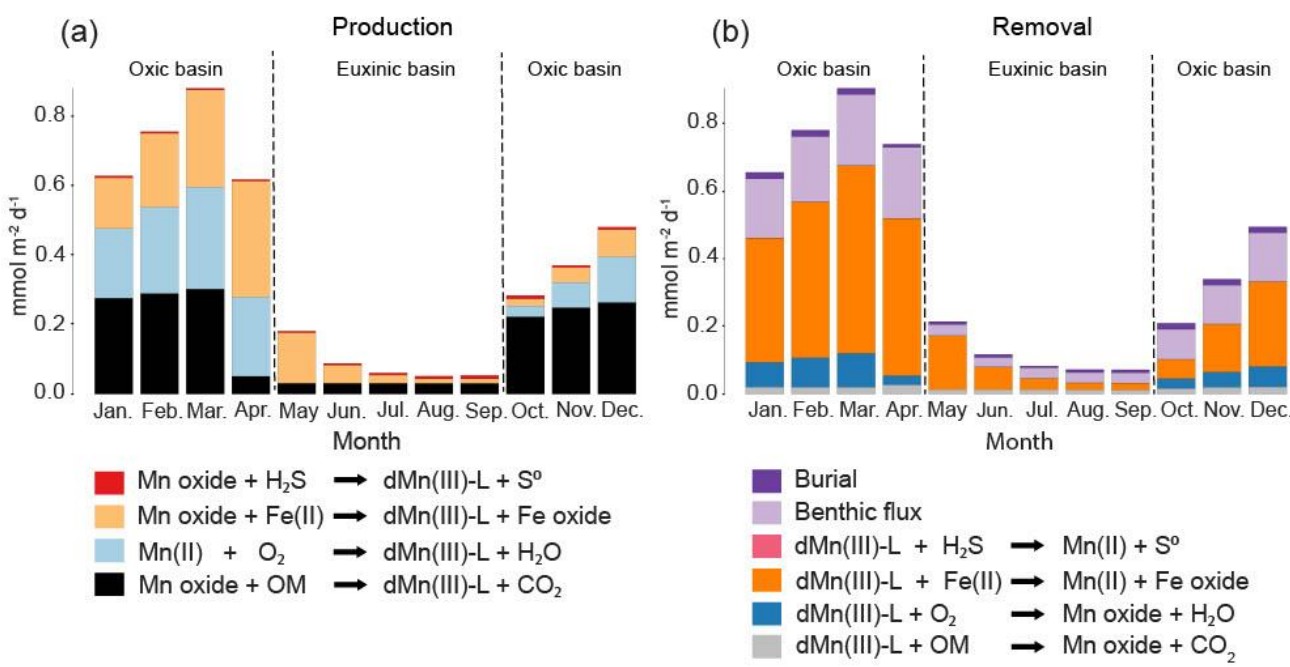

**Figure 6. Depth integrated reaction rates for the sediment processes that (a) produce dMn(III)-L and (b) that remove dMn(III)-L as calculated with the model. The rates are calculated for each month of one year (2020) and are given in mmol m$^{-2}$ d$^{-1}$.**

Our model also allows us to assess trends in depth-integrated rates of dMn(III)-L production and removal in the sediment throughout the year (Fig. 6; see Fig. S11 for modelled pore water profiles in the months between March and September 2020). Taking October as the starting point of the oxic period that lasts until April, we see that production of dMn(III)-L via reduction of Mn oxides coupled to OM degradation initially dominates. Over time, reduction of Mn oxides coupled to Fe(II)

oxidation and oxidation of dissolved Mn(II) by $O_2$ become increasingly important. Following the onset of anoxia in May reduction of Mn oxides by Fe(II) becomes the major source of dMn(III)-L. During the euxinic months, the role of reduction of Mn oxide by Fe(II) decreases and OM degradation becomes the major driver of Mn oxide reduction from August onwards, followed by $H_2S$ oxidation. Removal of dMn(III)-L from the sediment is dominated by reduction by dissolved Fe(II), especially during the oxic period, with additional loss through dMn(III)-L oxidation with $O_2$ (oxic period only) and benthic release.

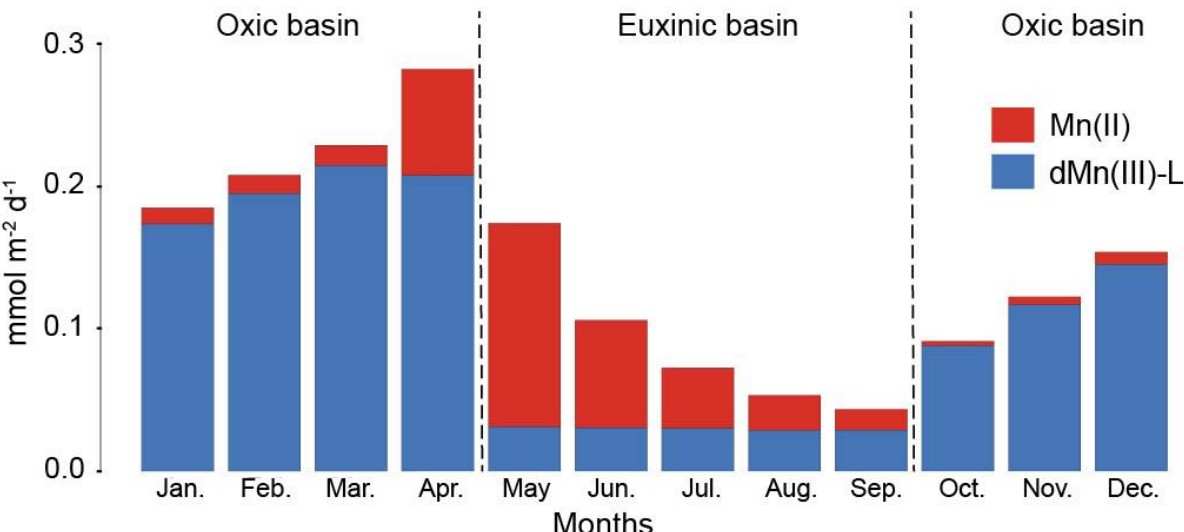

**Figure 7. Sediment-water exchange of dissolved Mn (mmol m$^{-2}$ d$^{-1}$) calculated per month in 2020. The total dissolved Mn flux consists of dissolved Mn(II) and dMn(III)-L. Positive numbers indicate a flux from the sediment to the overlying water.**

According to the model, the benthic release of dissolved Mn is highest in the period when the bottom waters are oxic (Fig. 7). During this time, the flux consists primarily of dMn(III)-L. Upon the onset of euxinia in May, the flux mainly consists of Mn(II). With time, the benthic flux of Mn(II) subsequently decreases allowing the relatively constant low flux of dMn(III)-L to gain relative importance.

### 3.3.3 Sensitivity analysis

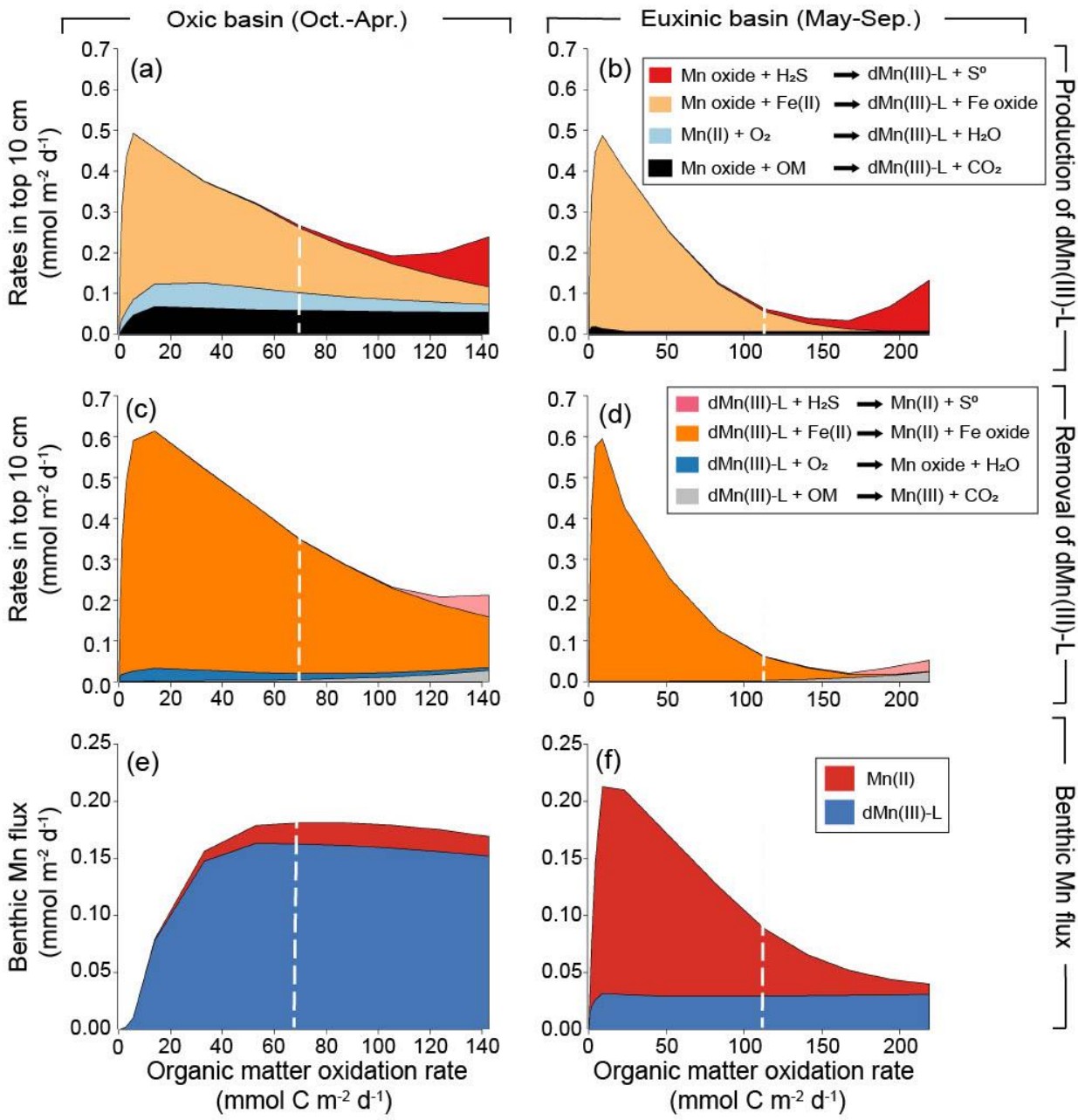

**Figure 8. Depth-integrated reaction rates (0 – 10 cm) of formation (a, b) and removal (c, d) of dMn(III)-L as a function of rates of organic matter oxidation in the sensitivity analysis. Note that the months when the basin is oxic (October – April; a, c) and euxinic (May – September; b, d) are separated. Average benthic flux of dissolved Mn(II) and dMn(III)-L as a function of the rate of organic matter oxidation in the months when the basin is oxic (e; October - April) and when the basin is euxinic (f; May - September). The vertical white dashed line represents the baseline run.**

To study the effect of variations in OM degradation rates on the benthic release of Mn, a model sensitivity analysis in which the input of OM was varied was performed. The sensitivity analysis revealed that, in the model, average rates of formation and removal of dMn(III)-L for the oxic and euxinic periods are highly dependent on OM oxidation rates (Fig. 8). Rates of all processes involving dMn(III)-L initially increase upon a rise in OM oxidation rates and then show a variable response, either decreasing (Fe(II), $O_2$), remaining largely constant (OM) or increasing ($H_2S$). Reactions involving dissolved Fe(II) generally dominate. The role of processes involving Fe(II) diminishes, however, during the euxinic months when rates of OM oxidation are above 150 mmol m$^{-2}$ d$^{-1}$.

According to the model, the average benthic flux of Mn during the oxic period is highly sensitive to OM oxidation rates between 0 and 50 mmol m$^{-2}$ d$^{-1}$ and shows a strong increase before stabilizing (Fig. 8). Most of the Mn is released as dMn(III)-L. The pattern is different for the euxinic period: here, again an initial increase in the Mn flux is observed when OM oxidation rates vary between 0 and 10 mmol m$^{-2}$ d$^{-1}$, but this is followed by a strong decrease. In this case, most of the Mn is released as Mn(II), except for the model runs with the highest OM oxidation rate: here dMn(III)-L gains importance.

### 3.3.4 Forward simulation

The forward simulation for 2021 using depth profiles of pore water $NH_4^+$, alkalinity, $SO_4^{2-}$, $H_2S$, TD Fe, TD Mn and sediment Mn oxides and Mn carbonates obtained for 2021 shows that, without any additional fitting, our model is able to capture the major trends in the sedimentary Mn cycle and key pore water constituents over a spring-summer-fall cycle (Fig. S12).

## 4 Discussion

### 4.1 Mn(III) is a key pore water component in a eutrophic coastal system

Dissolved Mn(III)-L is frequently observed in pore waters at sites where $O_2$ is present in the surface sediment (Madison et al., 2011, 2013; Oldham et al., 2019). Here, we show that dMn(III)-L is also a key pore water component in a setting where $O_2$ is absent and bottom waters are euxinic (Fig. 3). The results of our reactive transport model suggest that the pathways of production and removal of dMn(III)-L strongly depend on the pore water composition, with Fe(II) playing a critical role.

In March 2020, when bottom waters were oxic, a maximum in dMn(III)-L was observed in the top centimeter. This maximum is based on one individual data point, but we note that it is based on triplicate analyses (Fig. S5) and is in accordance with the peak in total dissolved Mn determined via an independent procedure (ICP-OES; Fig. S5). Additionally, such a sharp peak in dMn(III)-L at the oxic/anoxic interface is expected when $O_2$ is involved in the production of dMn(III)-L (Madison et al., 2013). The model underestimates the dMn(III)-L concentrations between depths of 10 to 20 cm, which we attribute to an incomplete understanding of the processes that impact dMn(III)-L production and formation in sulfidic

380 sediments. We note that the good fit of the model for most pore water and sediment components for 2020 and, as an outcome of the forward modelling, for 2021, gives confidence in the results.

According to the model, formation of dMn(III)-L in the pore water at our site in March is driven by - at equal rates - reduction of Mn oxides by OM and Fe(II) and oxidation of dissolved Mn(II) by $O_2$ (Fig. 5). This contrasts with the production pathways of dMn(III)-L estimated from pore water profiles and solid phase data for different sites in the Saint

385 Lawrence Estuary and Gulf, which pointed towards a dominant role for oxidation of Mn(II) by $O_2$ (Madison et al., 2013). This difference is likely due to the exceptionally large input of OM and Fe oxides at our study site, with a modelled average input of ~240 mmol C $m^{-2}$ $d^{-1}$ and ~19 mmol reactive Fe $m^{-2}$ $d^{-1}$ throughout the year (Fig. S1; approximately half of the reactive Fe is accounted for by Fe extracted in ascorbic acid Fig. S8, S9), compared to maximally 10 mmol C $m^{-2}$ $d^{-1}$ and 0.6 mmol Fe $m^{-2}$ $d^{-1}$ (Fe extracted with ascorbic acid) in the St. Lawrence Estuary and Gulf (Oldham et al., 2019).

390 In September 2020, when $O_2$ was absent from the pore water at our site, Mn oxide concentrations in the top layer of the sediment were low compared to March. This was likely the result of a lower input of Mn oxides when the bottom water was sulfidic, linked to quick reduction of Mn oxides in the water column. Under these conditions, dMn(III)-L was mainly formed via reduction of Mn oxides by OM, with a smaller contribution of Mn oxide reduction by $H_2S$ and Fe(II) (Fig. 5). Strikingly, dMn(III)-L co-occurred with $H_2S$ in the pore water, despite $H_2S$ concentrations of several millimolar (Fig. 3). A co-

395 occurrence of dMn(III)-L and $H_2S$ was observed previously in estuarine waters at $H_2S$ concentrations of several micromolar (Oldham et al., 2015). In this previous work, the co-occurrence was attributed to stabilization of dMn(III)-L by organic ligands that kinetically hindered Mn(III) reduction by $H_2S$ (Oldham et al., 2015). Our results suggest that stabilization of dMn(III)-L against reduction by $H_2S$ is even possible when $H_2S$ concentrations reach several millimolar.

According to our model, most dMn(III)-L at the study site is removed through reduction by Fe(II) (Fig. 5), explaining  the

400 strong counter gradients between dMn(III)-L and dissolved Fe in March (Fig. 3; Fig. S7). Apparently, the ligands that stabilize dissolved Mn(III) do not shield the Mn(III) against reduction by Fe(II). The larger role of Fe(II) as a reductant for dMn(III)-L then observed in the study by Madison et al. (2013) may be explained by the approximately 20 times higher Fe(II) concentrations at our site. Previous reactive transport modelling has highlighted the role of Fe(II) in the reduction of Mn oxides (Van Cappellen & Wang, 1996). What is novel here is that dissolved Fe(II) not only plays a key role in Mn oxide

405 reduction and dMn(III)-L production but also in dMn(III)-L removal. We note that this role of Fe(II) in dMn(III)-L dynamics came to a halt in August and September when the Fe-oxides that accumulated in winter were completely reduced and dissolved Fe precipitated with $H_2S$ as FeS and $FeS_2$. Taken together, this indicates that, in Fe rich coastal systems, sediment Fe dynamics can be an even more important driver of Mn cycling than previously considered (e.g. Madison et al., 2013). The results of the sensitivity analysis highlight that the role of Fe(II) in dMn(III)-L cycling holds over a wide range of OM

410 oxidation rates (Fig. 8).

Due to its organic complexation the diffusion coefficient of dissolved Mn(III) is expected to be lower compared to that of dissolved Mn(II) (Kalinichev and Kirkpatrick, 2007; Furukawa and Takahashi, 2008). Additionally, due to heterogeneity within the ligands that can stabilize dissolved Mn(III) (Madison et al., 2013; Oldham et al., 2015, 2017b), the diffusion

coefficient of the complex of dMn(III)-L can vary per location. We find that a difference in diffusion coefficient between dMn(III)-L and Mn(II) is essential to describe the sharp gradients in the dissolved Mn profiles in the model. Without a lower diffusion coefficient for dMn(III)-L it is not possible to form a sharp peak in dMn(III)-L near the sediment-water interface in our model (Fig. S13). Notably, all modelling studies of sedimentary Mn cycling to date focus only on dissolved Mn(II) excluding dMn(III)-L, which can explain why similar sharp gradients in dissolved Mn near the sediment water interface have so far been difficult to capture (e.g. Slomp et al., 1997; Reed et al., 2011). The adjusted diffusion coefficient for dMn(III)-L is 16.6 cm$^2$ yr$^{-1}$, ca. 8 times lower than the diffusion coefficient for Mn(II) (132.6 cm$^{-2}$ yr$^{-1}$; Table S6). When we assume all dissolved Mn is present as Mn(II) the calculated diffusive flux of dissolved Mn would be ca. 10 and 3 times higher than when we consider both Mn(II) and dMn(III)-L in March and September, respectively. The lower diffusion coefficient of dMn(III)-L when compared to Mn(II) also implies that calculated diffusive fluxes across the sediment-water interface will be overestimated if all Mn is assumed to be present as Mn(II) (Fig. S14).

## 4.2 Seasonality in benthic Mn fluxes

The model results imply that throughout the year, both dissolved Mn(II) and dMn(III)-L contribute to the release of Mn from the sediment to the overlying water (Fig. 7). The flux is highest and primarily consists of dMn(III)-L under oxic conditions in winter and spring, when the input of Mn oxides and recycling of Mn near the sediment-water interface is highest. The high contribution of dMn(III)-L to the benthic Mn flux in the oxygenated basin results from faster oxidation of Mn(II) with $O_2$ compared to dMn(III)-L, which leads to a build-up of dMn(III)-L just below the sediment-water interface and subsequently a high benthic flux of dMn(III)-L.

When euxinic bottom-water conditions are established in summer, the modelled benthic flux primarily consists of Mn(II), because Mn(II) is no longer oxidized by $O_2$ (Fig. 7). However, a fraction of the Mn released from the sediment remains dMn(III)-L, indicating that a part of the dMn(III)-L released is a product of Mn oxide reduction rather than Mn(II) oxidation. As sediments become depleted in Mn oxides, typically during persistent hypoxia or anoxia, the benthic flux of Mn is known to strongly decrease (Slomp et al., 1997; Lenstra et al., 2021a). We find that the benthic flux of Mn decreases substantially as soon as the basin becomes euxinic, which likely indicates that highly reactive Mn oxides are quickly removed from the sediment when the input of Mn oxides decreases as a result of the bottom water euxinia. This is supported by the pore water and sediment data for the fieldwork campaigns between March and October in 2021 (Supplements section 1; Fig. S12). The TD Mn in the pore water already decreases between March and April 2021 and remains low throughout the euxinic period that lasts from June to September (Żygadłowska et al., 2024b). We note that part of the dissolved Mn(II) in the pore water precipitates as Mn carbonate and hence is retained in the sediment. To visualize that both variations in Mn carbonate formation and input of Mn carbonate contribute to the seasonal variation in the sediment, a model run without Mn carbonate precipitation was performed (Fig. S15).

A new finding here is that, at the end of the anoxic period, the benthic Mn flux mainly consists of dMn(III)-L. The continuous release of dMn(III)-L from the sediment during the period of anoxia is reflected in the accumulation of dMn(III)-L in the anoxic part of the water column in September (Fig. 2). When bottom water $O_2$ re-establishes in October, the influx of Mn and Fe oxides, the rates of sedimentary Mn cycling and the benthic flux of Mn all increase. In subsequent months, the benthic flux of dMn(III)-L and the importance of Fe in Mn cycling continues to increase until just before the onset of a new euxinic period (Fig. 6, 7).

The sensitivity analysis reveals that the benthic fluxes of Mn(II) and dMn(III)-L are also dependent on the rate of OM oxidation in the sediment (Fig. 8). Nevertheless, the general patterns that emerged from the simulation for 2020, still hold: when bottom water $O_2$ is present, dMn(III)-L release from the sediment to the overlying water dominates over Mn(II) release. The reverse is found during the euxinic period when taken as a whole. Taken together, our results highlight that while Fe(II) dynamics play a critical role in dMn(III)-L production and removal in our Fe- and OM-rich coastal sediment, the presence of bottom water $O_2$ ultimately determines whether, on a yearly basis, Mn(II) or dMn(III)-L is the dominant form of Mn that escapes to the overlying water. Regardless of the bottom water redox conditions, the model always predicts some benthic release of dMn(III)-L.

## 4.3 Implications

Our results imply that dMn(III)-L should be considered as a potential pore water constituent when studying sedimentary Mn cycling in OM- and Fe-rich coastal sediments. When dMn(III)-L is released into anoxic waters, it can act as an oxidant and reductant, with potential implications for various redox interactions, as previously shown for anoxic water columns (e.g. Trouwborst et al., 2006; Yakushev et al., 2007; Oldham et al., 2015). When, in contrast, $O_2$ is present in the bottom water, dissolved Mn(II) will be readily oxidized to dMn(III)-L when it is exposed to $O_2$ near the sediment-water interface. As a consequence, dMn(III)-L is the main form in which dissolved Mn may leave the sediment (Fig. 7). The ligand stabilizing dMn(III)-L may not only slow down diffusion but is also likely to slow down the formation of solid phase Mn oxides (Sander & Koschinsky, 2011; Oldham et al., 2021). Because Mn oxides are subject to gravitational settling (e.g. Sulu-Gambari et al., 2017; Hermans et al., 2021) and dMn(III)-L is not, the dMn(III)-L is expected to be transported further away from the sedimentary source (Lenstra et al., 2020). Additionally, trace metals such as cobalt, nickel and zinc, can adsorb to the negatively charged surface of Mn oxides (Koschinsky and Hein, 2003). When transport of Mn is in the form of dMn(III)-L instead of Mn oxides, these trace metals will no longer adsorb to the Mn oxides and the transport of the trace metals will be decoupled from the transport of Mn (Oldham et al., 2021; Lenstra et al., 2022).

## Conclusions

Our combined seasonal field and modelling study reveals that dissolved Mn(III) (dMn(III)-L) is a key component of the Mn cycle in sediments of a seasonally euxinic coastal basin. Dissolved Mn(III) accounts for the majority of dissolved Mn in the pore water and coexists with high concentrations of $H_2S$. Results of a multicomponent reactive transport model suggest that, at our study site, reduction of Mn oxides coupled to oxidation of Fe(II) and organic matter and oxidation of Mn(II) by $O_2$ are the primary sources of the dMn(III)-L when $O_2$ is present near the sediment water interface. However, when the bottom waters are euxinic, reactions with dissolved Fe(II) likely dominate both the production and removal of dMn(III)-L. Modelled benthic Mn fluxes suggest that while dMn(III)-L is released from the sediment throughout the year, there is a distinct seasonal contrast: in the model, release of dMn(III)-L dominates when bottom waters are oxic while a transition to euxinia leads to a temporary increased role for Mn(II). The benthic release of dMn(III)-L, both under oxic and euxinic bottom waters, in combination with a higher mobility of dMn(III)-L in the water column when compared to Mn(II), may contribute to lateral transport of Mn from coastal zones to open marine waters.

## Code and data availability

The data presented in this paper are available in the supplements. The data and model presented in the study are also deposited in the Zenodo repository, doi: 10.5281/zenodo.11475150. These will be made public upon acceptance of the paper.

## Conflict of interest

The authors declare that they have no conflict of interest.

## Acknowledgements

We thank the captain and crew of the R/V Navicula, A. Tramper and M. Hermans for their support during the sampling campaigns. We are also grateful to T. Claessen, C. Mulder, J. Visser and H. de Waard, for analytical assistance at Utrecht University. This research was financially supported by ERC Synergy grant MARIX (854088) and the Netherlands Earth System Science Center (NESSC 024002001). WKL acknowledges support from a NESSC Travel Grant and Veni grant VI.Veni.222.332.

## Author contributions

RK, WL and CS designed the research and wrote the paper with comments provided by all authors. OZ, NvH, and WL performed the sampling. RK, OZ, NvH, and WL performed the geochemical analyses. VO instructed on the dissolved

Mn(II)/Mn(III) method. RK and WL wrote the reactive transport model code and performed the model simulations. RK, OZ, NvH, WL, CS and MJ interpreted the data. All authors contributed to the article and approved the submitted version.

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
