# Peer review of "Dissolved Mn(III) is a key redox intermediate in sediments of a seasonally euxinic coastal basin"

_EGUsphere, 2024_

## Author Comment (AC2)

[Figure]

Figure 1: Forward simulation of the model to data of NH4+, alkalinity, sulfate, H2S, dTFe, dTMn and, for 4 months, Mn oxides and Mn carbonates from the sampling events between March and October in 2021.

Figure 2: Model run where authigenic Mn carbonate precipitation is turned off (in red) versus the model base run (in black). The difference in Mn carbonate concentrations between the model without Mn carbonate precipitation and the base run indicate the amount of Mn carbonate that has formed within the sediment.

---

## Author Response (AR1)

*Dear Mr Klomp et al.,*

*Thanks for the responses to the comments and questions provided by two reviewers. I think the reviewers raise several good points, one suggesting minor, the other one major revisions. Based on these comments and my own review, I tend towards major revision. I understand how you plan to address their feedback, so now I look forward to your revised manuscript that incorporates suggestions from the reviewers and your responses. I consider sending your manuscript out for a second review.*

*Thank you and I look forward to your revised manuscript.*

*Kind regards,*
*Dr Sebastian Naeher*
*Associate Editor, Biogeosciences*

Reply:

Dear editor,

Thank you for your words. Below you find a point-by-point response to the reviews including a list of all relevant changes made in the manuscript. Below our response to the reviewers, there is a list of additional corrections made to correct typo's or to comply with the manuscript guidelines of Biogeosciences. All adjustments can be found in the revised manuscript and supplements. The track-changes file of the supplements can be found after the track-changes file of the manuscript, in the same file. The line numbers correspond with the clean text, without track-changes on.

Kind regards,
Robin Klomp
* * *
**Reply to Reviewer: 1**

*Comments:*
*Klomp et al. investigate seasonal differences in the sedimentary Mn cycling in a eutrophic coastal basin in the Netherlands. Their study provides a comprehensive data set and combines geochemical water column, pore-water and solid-phase data with reactive transport modeling.*

*The authors emphasize that dissolved Mn(III)-L is an important component of the Mn cycle throughout the year. Depending on the seasonal oxygen concentration in the bottom water and the influx of Mn oxides, Mn is predominantly released to the overlying water as dissolved Mn(III)-L in winter (oxic bottom water and higher influx of Mn oxides), whereas in summer (euxinic bottom water and lower influx of Mn oxides), both dissolved Mn(II) and dMn(III)-L are released, but Mn(II) is the dominant species. In contrast to dissolved Mn(II), the relatively stable and mobile dMn(III)-L may be transported from coastal areas into the open ocean. In addition, the biogeochemical processes leading to the formation and removal of dMn(III)-L in the sediment are strongly linked to the Fe cycle. Therefore, this study is very helpful in improving our understanding of sedimentary Mn and Fe cycling, especially the coupling between the two cycles, in coastal areas.*

*Overall, the manuscript is well written, however some of the figures could be improved (see specific comments). I would recommend the publication of this manuscript after some minor revisions.*

Reply: We thank the reviewer for the positive words and comments and suggestions. Below, we indicate how we addressed the comments in the revised manuscript.

*Specific comments*

*Abstract*

*Line 11: It is better to write "depth profiles of water, pore-water and solid-phase data" instead of just "depth profiles". Otherwise, it is not clear which comprehensive data set has been the basis for this study.*

Reply: We modified the text to comply with the reviewers comment.

**Revised text**: Line 13: "… of solutes and solids for the sediment and overlying waters …"

*Methods*

*Figure 1a: The map of Lake Grevelingen including the overview map of the Netherlands in the top right corner is too small. It is very difficult to identify the water depths, especially in the Scharendijke basin.*

Reply: We modified the figure as suggested.

A modified figure can be found in the revised manuscript Fig. 1.

*Results*

*Figure 3: In contrast to the water column profiles, the pore water profiles are relatively small. As some of the discussion relates specifically to the upper centimeters, it would be helpful if these were shown in more detail. In addition, the light grey dots and lines are hard to see.*

Reply: A zoom of the upper 20 cm of the porewater profiles is now added to the supplements. We changed the color of the light grey dots and lines to a different color.

A reference to the figure is added to the figure caption in line 227: "A zoom of the top 20 cm of the profiles is presented in Fig. 6.", and to the main text (lines 260 – 261): "…; for a zoom of the top 20 cm, see Fig. S6)".

A modified figure can be found in the revised manuscript (Fig. 3) and a zoom of the top 20 cm can be found in supplements Fig. S6.

*Figure 3h-j: If I see it correctly, the sum of dMn(III)-L and Mn(II) does not correspond to the TD Mn concentrations. Is there any explanation for this?*

Reply: TD Mn is measured independently of dMn(III)-L and Mn(II), therefore these values can differ slightly. Furthermore, the kinetic Mn(II)/Mn(III) method that we use does not include the Mn(III) that is bound to strong ligands, which are included in the ICP-OES results for TD Mn (Oldham et al., 2015). We assume that the difference between dMn(III)-L + dMn(II) and the TD Mn measured by ICP-OES is the fraction of dMn(III)-L that is bound to strong ligands. In the supplements figures S5 and S6 it is visible that in March TD Mn and the sum of dMn(III)-L and Mn(II) overlap well, but that in September TD Mn exceeds the sum of dMn(III)-L and Mn(II) in many measurements, indicating that the strongly bound dMn(III)-L makes up for a larger part of the dMn(III)-L in September. We revised the text to point this out.

Revised text (lines 160 – 162): "Strongly bound Mn(III)-ligand complexes cannot be measured via this method (Oldham et al., 2015; Kim et al., 2022). Therefore, the difference between the sum of measured Mn(II)/Mn(III)-L and the total dissolved Mn measured by ICP-OES can be interpreted as the amount of Mn(III)-L that is bound to strong ligands."

*Figure 4: Again, the light grey dots and lines are hard to see.*

Reply: The light grey color in the figure are changed to a different color.

A revised figure can be found in the revised manuscript Fig. 4.

*Line 237 + Figure 4a: As the Corg content usually refers to the total sediment weight, it is better to speak of Corg contents rather than Corg concentrations.*

Reply: We modified the text as suggested.

Revised text (line 288): "The $C_{org}$ content in the top 10 cm of the sediment was …"

*Line 245-311: The description of the model results already contains many possible interpretations or approaches for discussion. It would be good if these were directly linked with the discussion part.*

Reply: We agree that the points regarding the degassing and the seasonality in the solid phase profiles (line 246-251) could be interpreted as discussion. These lines are removed from the results. The remainder of the section describes the model results and does not include interpretation of the model results. We modified the text to clarify that we are presenting the outcome of the model.

Revised text:
Lines 292 – 293: The sentence: 'However, given degassing of $CH_4$ during sampling, we cannot capture actual $CH_4$ concentrations at this site (Żygadłowska et al., 2023a).' is removed. This is already covered in the methods, lines 228 - 230.

Lines 294 – 295: The sentence: 'This likely results from temporal variations in the reactivity and input flux of solid phases that are not included in the model (Table S4, S5)" is removed, because it covers a discussion about why the model fit is not perfect. This is now covered in the method section, lines 230 – 233: "… which also vary between years. This is especially true for the fit to the solid phase profiles since such variations in the reactivity and input flux of solid phases are not specifically included in the model (Table S4, S5)."

In line 299 we added: "…we will primarily focus on a description of the model results…"

*Figure 5: At first glance, the figure is a little confusing due to many numbers/rates. One suggestion here would be to split the figure in two: (a) situation in March, (b) situation in September. In this way, the size of the arrows could be adjusted to the respective rates to better highlight the differences in the rates between March and September. In addition, the oxygen conditions in the bottom water could then be integrated for both situations.*

Reply: We appreciate the suggestion of the reviewer but find that comparing the two seasons becomes difficult when the figure is split into two panels, because the outcome for the two seasons is then presented separate from each other. The seasonal contrast is a key aspect of this section of the discussion, therefore we prefer to keep the one panel figure. We added the redox state of the bottom water in both seasons in the panel as suggested.

A revised figure can be found in the revised manuscript Fig. 5.

*Discussion*

*Line 334: Again, it would be very good if the top 0 to 10 cm were shown in more detail to see the strong counter gradients.*

Reply: Please, see our reply to the earlier comment on Figure 3. A zoom of the top 20 cm of the porewater profiles is added to the supplements.

For revised figure, see supplements Fig. S6.

*Line 394-395: At this point it should be described in more detail to what extent the transport of trace metals is decoupled from that of Mn when it is mainly present as dMn(III)-L. Were trace metals measured in the water column samples that could confirm the mentioned hypothesis?*

Reply: We expanded the section of the text on the trace metals as suggested, referencing the relevant literature on this topic. Unfortunately, we do not have water column profiles of cobalt, nickel and zinc for March and September 2020 to include in this paper.

Revised text (Lines 471 – 474): "Additionally, trace metals such as cobalt, nickel and zinc, can adsorb to the negatively charged surface of Mn oxides (Koschinsky and Hein, 2003). When transport of Mn is in the form of dMn(III)-L instead of Mn oxides, these trace metals will no longer

adsorb to the Mn oxides and the transport of the trace metals will be decoupled from the transport of Mn (Oldham et al., 2021; Lenstra et al., 2022)."

**Technical corrections**

**Line 16: Since manganese (Mn) is introduced in line 11, oxygen should be introduced in the same way: Our model indicates that, when oxygen (O2) is present in the bottom water, there are three major sources of pore water dMn(III)-L (…).**

Reply : The text is modified as suggested.

Revised text (Line 17): "… when oxygen ($O_2$) is present in the bottom water, …"

*Line 43: Again, please also introduce oxygen for consistency.*

Reply : The text is modified as suggested.

Revised text (Line 48): "… oxygen ($O_2$) …"

*Line 307-308: Here the authors are certainly referring to Figure 8, as Figure 9 does not exist.*

Reply : We corrected this in the text.

Revised text (Line 358): "… (Fig. 8)."

*Line 370: The sentence structure is a bit awkward. It would be better: When bottom water O2 re-establish in October, the influx of Mn and Fe oxides, the rates of sedimentary Mn cycling and the benthic flux of Mn all increase.*

Reply : The text is modified as suggested.

Revised text (Line 448 – 449): "When bottom water $O_2$ re-establishes in October, the influx of Mn and Fe oxides, the rates of sedimentary Mn cycling and the benthic flux of Mn all increase."

*Line 388: Instead of saying "when it meets O2", it is more appropriate to write "when it is exposed to O2".*

Reply : The text is modified as suggested.

Textual revision (Line 466): "… when it is exposed to $O_2$ …"
* * *
**Reply to reviewer 2**

*General comment*

*The manuscript written by Klomp et al. describes the manganese cycle in a coastal sediment with extremely high accumulation rates (> 10 cm yr$^{-1}$) and overlaid by a seasonally anoxic saline water. The manuscript recycles part of the data published by Żygadłowska et al. (2023) with the addition of new data on manganese speciation (called Mn(II) and Mn(III)=L). The main originality of the manuscript is the use of dissolved manganese speciation measurements into a reactive transport model.*

*Most of the kinetic parameters concerning Mn(III)=L are deduced from the model fit, which allows the author to discuss the reactivity of Mn(III)=L and its importance in manganese efflux from the sediment. Beside the clear importance of this topic, the real novelty of this approach for Mn speciation and the good quality of the dataset, it seems that in the discussion, the authors feel too confident about the model and the analytical results and avoid discussing the underlying hypothesis and limitations. Ultimately, it leads to a general overinterpretation of the data with many direct affirmative sentences not supported by detailed argumentation. Model results are taken as true while they rely on many cases of hypotheses hidden by the complexity of the model, preventing the reader to appreciate the model's limitations and thus its scope.*

*In particular,*

*a) the model fit to Mn(III)=L is not properly discussed while it fits mainly to one unique Mn(III)=L measurement (March, 0-1 cm depth) and fails to fit the deeper part of the Mn(III)=L profile;*

*b) the analytical demonstration of the true occurrence of Mn(III)=L is not detailed while caution have been published on this method since the Madison et al. (2011) paper (Kim et al., 2022) which requires a particular effort of clarity;*

*c) it seems that most of the model output are not produced by the modelled chemical reaction but mainly result from the model input i. e. by the strong seasonality of the manganese oxides deposition rate chosen by the authors but not discussed;*

*d) Important parts of the sedimentary Mn cycle are not discussed, neither mentioned, in particular the interaction with the nitrogen cycle and the role of adsorbed Mn$^{2+}$. These reservations are detailed below.*

Reply: We thank the reviewer for the thorough assessment of our manuscript and for the positive words regarding the novelty of our approach and the quality of the data set. We have done our best to address all comments below and have revised the manuscript accordingly.

Regarding the points raised above, we note that the paper of Żygadłowska et al. (2023) focusses on pathways of methane removal at this site, with the Mn data provided only to allow an assessment of the role of Mn oxides as a potential electron acceptor in methane oxidation. We expanded the model sections to clarify the goals, the assumptions made, the limitations of the model and the boundary conditions. We expanded the section on the analysis of Mn(III)-L. We also added a section discussing the potential for interactions with the nitrogen cycle and why we can exclude a major role for adsorbed Mn(II). We modified the text in many places to clarify that these are model results, see directly below. Further details are provided after each individual comment further below.

Text revised to emphasize the use of a reactive transport model:

Line 15: "Our model results suggest"…"

Line 16: "According to the model …"

Line 19: "Removal of pore water dMn(III)-L is inferred to …"

Line 20: "… model-calculated rates …"

Line 22: "Modelled benthic release…"

Line 24: "Our model findings highlight …"

Line 76: "Our model results suggest …"

Line 228: "In the model, high rates…"

Line 306: "… show that, according to the model, the …"

Line 336: "According to the model, the …"

Lines 350 – 351: "… a model sensitivity analysis in which the input of OM was varied was performed."

Line 251: "revealed that, in the model, …"

Line 357: "According to the model, the …"

Line 372: "The results of our reactive transport model suggest that …"

Line 382: "According to the model, formation of dMn(III)-L …"

Line 299: "According to our model, most dMn(III)-L at the study site is …"

Line 320: "… as calculated with the model."

Line 479: "Results of a multicomponent reactive transport model suggest …"

Line 482: "reactions with dissolved Fe(II) likely dominate…"

Lines 482 – 483: "Modelled benthic Mn fluxes suggest that …"

Line 484: "in the model, release of dMn(III)-L …"

**Main reservations**

*a) The model fit to Mn(III)=L and Mn(II) seems insufficient to deduce fluxes, production and constant rates with a high level of confidence. First, it seems very dangerous to base most of the*

*paper interpretation only on one unique Mn(III)=L (March, 0-1 cm depth) measurement since contamination or analytical errors are always possible.*

Reply: Indeed, the peak in Mn(III)-L near the sediment-surface in March is based on one observation only, and we now specifically note this in the revised text and discuss the related uncertainty. We indicate that this was the result of triplicate analyses and that the total dissolved Mn concentration, which was determined in an independent procedure, also showed a peak of a similar magnitude at this depth (visible in Figs. S5 and S6, which is now used to illustrate this). Importantly, a very sharp peak in Mn(III) is expected, because the major pathway to produce Mn(III) is oxidation of Mn(II) by $O_2$. Since $O_2$ is only present in the upper 0.6 cm of the sediment (Fig. 3), Mn(III) is expected to be mainly produced in the upper 0.6 cm. Our observations are also in line with other studies on Mn(III)-L in sediments showing a sharp peak of Mn(III)-L near the sediment-surface, recorded in only limited data points (e.g. Madison et al., 2013). Notably, the vertical redox zonation at our site is more compressed than in this previous work, explaining the sharper peak.

Revised text (Lines 364 – 369): "…were oxic, a maximum in dMn(III)-L was observed in the top centimeter. This maximum is based on one individual data point, but we note that it is based on triplicate analyses (Fig. S4) and is in accordance with the peak in total dissolved Mn determined via an independent procedure (ICP-OES; Fig. S4). Additionally, such a sharp peak in dMn(III)-L at the oxic/anoxic interface is expected when $O_2$ is involved in the production of dMn(III)-L (Madison et al., 2013)."

*Even is the data is validated, the sampling uncertainty on one point should obviously produce important uncertainties in the model results. For example, the authors suppose that the observed maximum is at 0.5 cm depth, while it could also occur at 0.2 cm, given the centimeters resolution of the sampling. Does such difference significantly change the model results ? Many additional sampling uncertainties described in the literature should prevent the author to stand most of their results on only one measurement (spatial heterogeneity from macro organism, erosion during sampling with a gravimetric core, loss of any fluffy layer on the top of the sediment, …).*

Reply: The goal of the modeling is to obtain insight in the main drivers of trends in sedimentary Mn cycling in a coastal system that is seasonally euxinic. We now expanded the text to explain that limitations related to our sample resolution near the sediment surface do not alter the overall picture of Mn cycling at this location. We note that macrofauna are absent at this site and that our sampling method allows the fluffy layer on top of the sediment to be preserved. This is added in the text.

Textual revision (line 104 - 105): "Macrofauna are absent from the sediment in Scharendijke basin (based on visual observations of sediment sieved over 0.5 mm)."

Textual revision (line 121): "During sampling, the surface sediment remained intact."

Textual revision (line 2224 – 227): "Here we specifically focus on the general trends in the key processes that regulate the seasonal dMn(III)-L dynamics in a eutrophic basin. As the model focusses on these general trends, the overall picture of Mn dynamics at this site will not change due to uncertainties related to, for example, the sample resolution in the top part of the sediment."

*Second, the fit favors the Mn(III) maximum in the 0-1 cm depth layer, at the cost of a bad fit at depth. Could it be possible to ignore the high value at the top to favor a good fit at depth ? What would be the model result in this case ? Why do you not select these results?*

Reply: The processes that form the Mn(III) peak in the 0–1 cm depth layer are not coupled to those that control the Mn(III)-L profile deeper in the sediment. In the model, the presence of $O_2$ predominantly controls the formation of Mn(III)-L in the surface sediment, while at depth dMn(III)-L is mainly formed via interactions of Mn oxides with Fe and $H_2S$. Therefore, our good fit at the top does not directly impact the Mn model fit deeper in the sediment. We agree with the reviewer that we do not fully capture the Mn(III)-L profile at depth. This is likely because of an incomplete understanding of the processes that impact Mn(III)-L production and formation in anoxic/sulfidic sediments. We added text to explain this in the revised manuscript.

Textual revision (lines 297 -298): "… exceptions are the modelled $CH_4$ profile and the profile of dMn(III)-L between 10 and 20 cm."

Textual revision (line 378 – 380): "The model underestimates the dMn(III)-L concentrations between depths of 10 to 20 cm, which we attribute to an incomplete understanding of the processes that impact dMn(III)-L production and formation in sulfidic sediments."

Textual revision (line 415 - 417): "Without a lower diffusion coefficient for dMn(III)-L it is not possible to form a sharp peak in dMn(III)-L near the sediment-water interface in our model (Fig. S12)."

*b) The identification of Mn(III)=L needs to be strengthened since skeptical points of view have been published about this method (Kim et al., 2022). The credibility of the competitive ligand exchange kinetic methods requires more information about the deconvoluting of the kinetic signal and the precise conditions of the essay. In particular, the kinetic of manganese complexation is very sensitive to the chlorite content during the measurement, as detailed in (Thibault de Chanvalon and Luther, 2019). The oxygen concentration during the essay is also critical and should be discussed (Kim et al., 2022). I recommend publishing as supplementary material some examples of the time series of Mn=porphyrin formation rate including the March 0-1 cm depth sample, together with detailed essay conditions (salinity, oxygen), the strategy developed to overcome the method known limitations and the profile of apparent rate constants obtained for Mn(II) and Mn(III)=L.*

Reply: To correct for the salt dependence of the kinetics of the Mn(II) – porphyrin reaction, the k-value for the reaction between Mn(II) and porphyrin was determined in triplicate in an aliquot of a sample, which was completely reduced by adding hydroxylamine for 24 hours, as proposed in Oldham et al. (2015). This is now mentioned in the revised manuscript. Graphs of the kinetic lines used to determine the k1 value are added to the supplements of the paper. In addition to that, graphs of several kinetic curves of samples are included in the supplements.

Regarding the oxygen concentrations: we followed the protocol of Madison et al. (2011) which allows for its presence during the measurement. During all measurements, the oxygen concentrations were the same, so an effect of $O_2$ on the kinetic measurements would be similar in all analyses and therefore would not cause a difference between the measurements that could result in a drift in the outcome. Oxygen mainly affects Mn(III) bound to strong ligands (e.g. DFOB; Kim et al., 2022). It is already known that the method we use does not effectively target Mn(III) bound to strong ligands like DFOB (Madison et al., 2011; Oldham et al., 2015). Therefore, we do not expect our measurements to be affected by $O_2$. We now explicitly mention in the manuscript that $O_2$ was present during the measurements and that this affects the measurement of strong ligands but not those targeted here.

Textual revision (line 152 – 162): "Dissolved Mn(II) and dMn(III)-L concentrations were determined simultaneously via a kinetic spectrophotometric method using porphyrin, cadmium chloride and an imidazole borate buffer as described previously (Madison et al. 2011; detection limit of 1 $\mu$mol $L^{-1}$

for Mn). The kinetics of the Mn(II) – porphyrin reaction depends on environmental characteristics such as salinity (Thibault de Chanvalon and Luther, 2019) and should therefore be determined for each site separately. Here, the kinetic constant value for Mn(II) was determined in triplicate on an aliquot of sample in which the dissolved Mn was completely reduced by adding an excess of hydroxylamine (Oldham et al., 2015; for kinetic curves of the triplicate analysis see Fig. S1). All samples were analyzed in triplicate using a 1 cm pathlength quartz cuvette in a Shimadzu UV-1900i spectrophotometer (for examples of the kinetic curves, see Fig. S2). Our analyses were all carried out under normal atmospheric conditions. Strongly bound Mn(III)-ligand complexes cannot be measured via this method (Oldham et al., 2015; Kim et al., 2022). Therefore, the difference between the sum of measured Mn(II)/Mn(III)-L and the total dissolved Mn measured by ICP-OES can be interpreted as the amount of Mn(III)-L that is bound to strong ligands."

Figures of the kinetic curves to determine the k1 value and kinetic curves for several samples can be found in the supplements of the revised manuscript, figures S1, S2.

*c) Most of the model output is not produced by the modeled chemical reaction, but by the model input: it seems that the Mn-ox concentration in the settling particles varies from 9.6 µmol/g in winter to 0.4 µmol/g during euxinic condition. Such important forcing needs to be discussed in detail, along with the most important geochemical reaction constraining the system. In particular, 1 - Zygadlowska et al. 2023 measured suspended material concentration and demonstrated that the bottom Mn concentration in particles does not change so much between seasons;*

Reply: The model output is determined by the chemical reactions and the boundary conditions that are assumed. As indicated in the detailed model description in the supplement (section 2), the input flux of metal oxides was set by fitting the model profiles to the measured profiles. This is the common procedure used in diagenetic modeling (Berg et al., 2003; Reed et al., 2016; Van Cappellen and Wang, 2018) because settling rates of metal oxides from the water column are very difficult to determine accurately. Indeed, Żygadłowska et al. (2023) measured the concentrations of suspended particulate Mn in the water column at 43 m water depth (i.e. 2 meters above the sediment-water interface) but without the settling velocity of the particles, this cannot be translated to a Mn input flux. Moreover, there is likely also lateral transfer of particles near the sediment-water interface along the slopes of this relatively small basin. We now added this in the text.

To address the concerns of the reviewer we also performed a forward simulation of the model using a data set for a range of porewater components and sediment Mn oxide and Mn carbonate for 8 sampling events in 2021, capturing the period between March and October. While we do not have dMn(III)-L and Mn(II) data for 2021, we do have detailed profiles of $NH_4^+$, alkalinity, sulfate, $H_2S$, TD Fe, TD Mn and, for 4 months, Mn oxides and Mn carbonates. The results (Fig. S11 in the revised supplements) illustrate that, without any adjustment, our model is able to capture the major trends in the sedimentary Mn cycle and key porewater constituents over a spring-summer-fall cycle. This gives confidence in the results of the model. The data-model comparison for 2021 (see figure S12 in revised supplements) also supports our finding for 2020 that variations in Mn oxide profiles are strongly impacted by the input of Mn oxides. If the decrease in Mn oxide concentrations in summer would be due to removal by chemical reactions only, an increase in dissolved Mn would be expected at the start of the euxinic conditions. We see, however, that when euxinia establishes in the bottom water (which in 2021 occurred in June; Zygadlowska et al., 2024b), total dissolved Mn concentrations were already decreasing. A short description of the fieldwork campaign performed in 2021 and the model results and data for 2021 are added to the supplements.

Revised text (lines 98 – 100): "Scharendijke basin is a relatively narrow and deep basin in an overall shallow lake (Figure 1). As a consequence, vertical transport settling of suspended matter is expected to be supplemented by lateral transport of material from shallower areas near the sediment-water interface."

Revised text (lines 107 – 108): "Additional sampling was carried out in eight field campaigns between March and September 2021, as described in the supplement (Section 1)."

Revised text (line 219): "The model was fit to the data set for 2020."

Revised text (lines 239 – 240): "A forward simulation of pore water and solid phase data collected during 8 sampling events, one in each month from March to October 2021, was performed to verify the model."

Revised text (lines 363 – 367): "3.4.4 Forward simulation

The forward simulation for 2021 using depth profiles of pore water $NH_4^+$, alkalinity, $SO_4^{2-}$, $H_2S$, TD Fe, TD Mn and sediment Mn oxides and Mn carbonates obtained for 2021 shows that, without any additional fitting, our model is able to capture the major trends in the sedimentary Mn cycle and key pore water constituents over a spring-summer-fall cycle (Fig. S11)."

Revised text (lines 371 – 376): "We note that the good fit of the model for most pore water and sediment components for 2020 and, as an outcome of the forward modelling, for 2021, gives confidence in the results."

Revised text (Supplements lines 55 - 72):

"1.      Description of the fieldwork campaigns performed in 2021

Sampling of the sediment at Scharendijke basin was performed during 8 sampling campaigns with RV Navicula each month from March to October 2021, following the same methods used in the fieldwork campaigns in 2020 presented in this study. An additional sampling event in August / September 2021 is reported in Żygadłowska et al. (2024a, b). These results are not included here, because we focus on the monthly output of the reactive transport model. During each campaign, a sediment core was collected using a UWITEC corer with a transparent PVC core liner (120 cm length, 6 cm inner diameter) to collect samples for $NH_4^+$, alkalinity, $SO_4^{2-}$, $H_2S$, TD Fe, TD Mn and during four campaigns, for the determination of Mn oxide and Mn carbonate. The core was sectioned at a 1 cm resolution under a $N_2$ environment, the sections were subsequently centrifuged to separate the pore water from the solid phase. The supernatant was filtered over 0.45 µm pore size filters. Samples for the analysis of $NH_4^+$ were stored at -20°C and later analyzed using the indophenol blue method (Solórzano, 1968). The samples for alkalinity and $SO_4^{2-}$ were stored at 4°C until analysis. Alkalinity was measured through titration with 0.01 M HCl, within 24 h after sampling and $SO_4^{2-}$ was measured using ion chromatography (Metrohm 930 Compact IC Flex; detection limit for $SO_4^{2-}$ of 50 µmol L$^{-1}$). The samples for $H_2S$ analysis were diluted five times in a 2% Zn-acetate solution in a glass vial and stored at 4°C. These samples were analyzed using the phenylenediamine and ferric chloride method (Cline, 1969). Samples for TD Fe and Mn were acidified with 10 µL 35% suprapure HCl per ml of sample and stored at 4°C. These samples were analyzed via ICP-OES (Perkin Elmer Avio; detection limit 0.1 µmol L-1 and 0.03 µmol L$^{-1}$ for Fe and Mn respectively). Samples for the analysis of the sediment residues were stored in $N_2$ purged aluminum bags at -20°C. Later, the samples were freeze-dried and ground with an agate mortar and pestle under a $N_2$ environment. Solid phase

speciation of Mn was then determined using a sequential extraction procedure as described in Lenstra et al. (2021)."

A figure with the comparison of the fieldwork data from 2021 with the computer model can be found in supplements Figure S12.

To clarify the separation of the computer model fitted to the data obtained in 2020 and the forward simulation using the dataset obtained in 2021, an indication of the years 2020 is added to lines 104, 120, 260, 289, 297, 300, 306, 312, 324, 334, 374 and 390.

*If important Mn oxide consumption in euxinic water is credible (e. g. Thibault De Chanvalon et al., 2023) why should it be the same for Mn carbonate? I was expecting an increase of Mn-carbonate in this case as it occurs in the euxinic sediment and because primary production favors carbonate precipitation;*

Reply: The variations in the profile of Mn carbonate in the model are assumed to be the combined result of variations in authigenic Mn carbonate formation and the input of Mn carbonate at the sediment-water interface. As described above for the Mn oxides, the input flux of Mn carbonates at the sediment-water interface was obtained by fitting the model to the data. Importantly, we could not fit the model to the data when assuming that authigenic Mn carbonate formation was responsible for all of the variation in the Mn carbonate profiles.

The relative roles of the two processes can be made visible by turning off authigenic Mn carbonate formation in the model (see the figure below). Importantly, the observed oscillations are too large to be solely explained by authigenic Mn carbonate formation, as this would require a higher input of Mn oxides and higher concentrations of dissolved Mn(II) concentrations in the sediment than observed. The approach, results and uncertainties in the Mn carbonate modelling are explained more extensively in the revised manuscript. We also included a figure illustrating this in the supplements.

Revised text (Lines 428 − 430): "We note that part of the dissolved Mn(II) in the pore water precipitates as Mn carbonate and hence is retained in the sediment. To visualize that both variations in Mn carbonate formation and input of Mn carbonate contribute to the seasonal variation in the sediment, a model run without Mn carbonate precipitation was performed (Fig. S14)."

Revised text (supplements lines 58): "… the influx of Fe oxides, Mn oxides, Mn carbonates and OM and the sedimentation rate (Fig. S3) "

Revised text (supplements lines 63 - 64): "The input of Mn carbonates was also varied to fit the sediment profiles, because authigenic Mn carbonate formation alone was insufficient to induce the observed oscillations."

*There is no direct proof of H2S in September; 4 – why assuming that anoxic water is necessary euxinic while transitory period with dominance of dissolved manganese has been observed over months in similar environment (Shaw et al., 1994)?*

Reply: The measured bottom water concentration of $H_2S$ in September 2020 was 111 µmol $L^{-1}$, which is direct proof of $H_2S$ in the water column. This value was indicated in the supplement (Table S7) and the data point is included in Figure 3 and the data file. We now emphasized this in the main text i.e.

emphasized the proof that the bottom waters were euxinic in 2020. Furthermore, we can deduce the presence of $H_2S$ in the bottom water in September 2020 from the drawdown of Mo in the bottom waters, as described in Zygadlowska et al. (2023 and 2024a) and from the seasonal accumulation of Mo in the sediment as described by Egger et al. (2016). The presence of $H_2S$ in the bottom water in summer was also observed in 2021 as described in Zygadlowska et al. (2024b). We updated the reference and specifically mention this evidence for recurring euxinia in the revised manuscript.

Revised text (lines 94 – 98): "… anoxic and sulfidic bottom waters, as recorded by the seasonal drawdown of molybdenum from the water column and its consequent sedimentary enrichment (Egger et al., 2016; Żygadłowska et al., 2023, 2024b). Water column euxinia was confirmed by direct measurements of $H_2S$ in 2021 (Żygadłowska et al., 2024a, b). Each year, the water column mixes again in autumn, resulting in bottom water reoxygenation."

***The discussion should clearly underline that most of the seasonality is driven by settling particle composition, while it is currently suggested by the topic discussed in section 4.2 that the "sediment becomes depleted in Mn oxides" because of sediment efflux and OM oxidation.***

Reply: We mention that the seasonality is mainly driven by the input of Mn oxides in section 4.2, lines 381 – 382: "The flux is highest and primarily consists of dMn(III)-L under oxic conditions in winter and spring, when the input of Mn oxides and recycling of Mn near the sediment-water interface is highest. We placed more emphasis on this seasonal variation driven by the variation in input in the revised manuscript.

Revised text (lines 448 – 449): "When bottom water $O_2$ re-establishes in October, the influx of Mn and Fe oxides, the rates of sedimentary Mn cycling and the benthic flux of Mn all increase."

Revised text (lines 439): "… when the input of Mn oxides decreases as a result of the bottom water euxinia."

***Nice oscillations for Mn-carbonate and Corg content in the sediment are reported and present phase opposition (maximum on one fit with the minimum of the other); is there any possibility to explain them because of geochemical reaction rather than because of input seasonality ?***

Reply: Please, see our previous reply regarding the oscillations in Mn carbonate formation and the figure above. We note that the degradation of organic matter in the sediment is strongly constrained by the ammonium profiles which show only modest variations between the different months in 2020 and 2021. This implies that we are capturing the seasonal changes in organic matter degradation. In this case, we can only model the variations in $C_{org}$ burial by invoking a variation in input. Such a variation is in line with variations in primary productivity and organic matter supply from the North Sea known for this system (e.g. Hagens et al., 2015). We clarified this in the text.

Revised text (supplements lines 32 - 33): "Hence, the rate of OM degradation in the sediment is directly linked to the $NH_4^+$ profiles."

Revised text (supplements lines 66 – 67): "… because the effect of a spring bloom was simulated. Such variations in OM input are in line with variations in primary productivity and OM supply from the North Sea known for this system (Hagens et al., 2015)."

*Something in the model switches the oscillation from phase opposition to in-phase oscillation below 65 cm depth, what it is ?*

Reply: Indeed, there is such a switch at 65 cm depth. Because it is visible in the data, we also changed the forcing in the model by changing the boundary conditions for the input of Mn oxides, $C_{org}$ and the sedimentation rate at this time in the simulation. We do not completely understand the mechanisms that drive this change in oscillation phase at depth, although we do see that this is a zone of increased sediment compaction. This switch does not change our findings regarding the Mn cycle, however, because the active cycling of Mn occurs mainly in the upper 25 cm of the sediment (see Fig. S6) and the change referred to occurs below 65 cm.

Revised text (supplements lines 68 – 70): "Between 60 – 80 cm depth, which is around where the shift in sedimentation rates is assumed, a shift in the oscillations is visible in for example the $C_{org}$ and Mn oxides sediment profiles (Fig. 4). This shift is modelled by varying the input of these compounds through the seasons."

*The observed loss of 4 umol/g of Mn oxides between March and September in the top 5 cm sediment would require a sediment efflux of approximately 4 umol/g x 2 600 g/dm³ x (1-0.90) x 0.5 dm / 6 months = 87 umol/dm2/month = 290 umol/m2/d which approximately fits with the observed gradient and the calculated flux of 210 umol/m2/d in March. So, on one hand, the rapid calculation suggests biogeochemical processes strong enough to produce the observed MnO2 depletion between March uand September. And on the other hand, the elaborated model requires very strong forcing in the Mn content input to fit the data. Is the rapid calculation I propose wrong ? why ?*

Reply: We remind the reviewer of the exceptionally high sedimentation rate of around 20 cm per year at this site. This implies that the top 5 cm of the sediment has been deposited in approximately 3 months instead of the 6 months assumed in the calculation. This would mean a doubling of the flux to 580 µmol m$^{-2}$ d$^{-1}$, which no longer fits with any of the fluxes calculated from the data.

The rapid sedimentation rate also implies that sediment that was at the sediment-water interface in March is located at a depth of around 10 cm in September. The concentration of Mn oxide at the sediment surface in March is comparable to the concentration at 10 cm in September, both around 4.5 µmol g$^{-1}$, which means that, even when the removal processes in March would be strong enough to remove 4 µmol g$^{-1}$ Mn oxide, such strong removal is not observed in the data.

*Why does the published model decrease the benthic efflux as soon as the water column becomes anoxic? I expected the opposite, the absence of oxygen should favor Mn efflux (since there is no more MnO2 precipitating in the oxygenated layer) until reactive MnO2 is consumed (which should take about 6 months). Why is it not modelled ? This counterintuitive result should be underlined and discussed.*

Reply: The benthic flux of dissolved Mn decreases relatively fast upon the occurrence of bottom water euxinia, because the highly reactive Mn oxides in this system are removed very quickly. As a consequence, there is not much highly reactive Mn oxide remaining when the system becomes euxinic. Therefore, the increase in benthic flux of Mn due to the release of dissolved Mn from Mn oxides that are reduced when the system becomes euxinic is smaller than expected. This is confirmed by the forward modeling for 2021 (see the figure S12), where total dissolved Mn in the porewater decreases

with a factor of around 4 from March to April and remains low throughout the euxinic period that lasted from June to September in that year. As indicated above, these data and the model fits are now included in the supplements.

Revised text (lines 434 – 449): "We find that the benthic flux of Mn decreases substantially as soon as the basin becomes euxinic, which likely indicates that highly reactive Mn oxides are quickly removed from the sediment when the input of Mn oxides decreases as a result of the bottom water euxinia. This is supported by the pore water and sediment data for the fieldwork campaigns between March and October in 2021 (Supplements section 1; Fig. S11). The TD Mn in the pore water already decreases between March and April 2021 and remains low throughout the euxinic period that lasts from June to September (Żygadłowska et al., 2024b)."

*I also suggest comparing the sedimentary Mn efflux taking into account Mn(III)=L (figure 7) with those calculated without Mn(III)=L as probably done in Żygadłowska et al., 2023.*

Reply: As suggested by the reviewer, we calculated the benthic fluxes based on the porewater profiles assuming the diffusion coefficient for Mn(II). In March, the benthic flux calculated without correcting for the Mn(III)-L concentration is about 10x larger than the modelled flux where Mn(III)-L is taken into account. In September, the calculated benthic flux is about 3 times larger than the modelled flux. These calculations are now incorporated in the methods and discussion.

Text as revised (line 195 - 204):

**"2.6 Calculation of benthic diffusive fluxes**

Diffusive fluxes of dissolved Mn across the sediment-water interface were calculated with Fick's law of diffusion, based on the gradient in total Mn concentration between the bottom water and the pore water in the upper cm of the sediment (at an average depth of 0.5 cm) by applying the formula:

$$J = -\varphi D_s \frac{dC}{dz} \qquad (1)$$

where J is the diffusive flux in mmol m$^{-2}$ d$^{-1}$, φ is the porosity of the sediment, Ds is the diffusion coefficient for Mn in the sediment in m$^{-2}$ d$^{-1}$, C is the concentration of Mn in μmol m$^{-3}$ and z is the sediment depth in m. In our calculations, we assumed all Mn was present as Mn(II), The Ds for Mn(II) was corrected for temperature and salinity using the R package CRAN: marelac (Soetaert et al., 2010), taking into account the tortuosity of the sediment (Boudreau, 1996)."

Revised text (Lines 277 – 278): "The diffusive benthic Mn fluxes calculated with Fick's law of diffusion, based on the concentrations of the total dissolved Mn, were 2.1 mmol m$^{-2}$ d$^{-1}$ in March and 0.09 mmol m$^{-2}$ d$^{-1}$ in September."

Revised text (Lines 420 – 422): "When we assume all dissolved Mn is present as Mn(II) the calculated diffusive flux of dissolved Mn would be ca. 10 and 3 times higher than when we consider both Mn(II) and dMn(III)-L in March and September, respectively."

*d) Some known reactions important in the sedimentary Mn cycle are not modeled or discussed. The revised manuscript should explain and justify why it seems negligible in your site. For example, Mn(III)=L oxidation by nitrite studies (Luther et al., 1997; Luther et al., 2021; Karolewski et al., 2020); debates on Mn-annamox (Hulth et al., 1999; Thamdrup and Dalsgaard, 2000); or the role of adsorbed Mn2+ (Richard et al., 2013; van der Zee et al., 2001; Canfield et al., 1993).*

Reply: Explanations for the exclusion of the mentioned processes in the model are now included in the revised model description in the supplements. We have the following argumentation:

We measured nitrate and nitrite concentrations in the upper 10 cm of the sediment in September 2020 with a Gallery™ Automated Chemistry Analyzer type 861 (Thermo Fisher Scientific). Nitrite and nitrite concentrations did not exceed the detection limit. Given these results, we do not expect an effect of interactions of nitrite with Mn cycling at this site. The nitrite and nitrite data are included in the results section and we now explain why the processes mentioned were not included in the model.

The relevance of Mn-annamox in marine environments is still under debate, as pointed out in the two references given (Hulth et al., 1999; Thamdrup and Dalsgaard, 2000). Because we are not certain whether this process can take place, we did not include it in the reactive transport model. We now mention this in the revised model description.

The primary effect of sorption of dissolved Mn(II) on Mn cycling is the enhanced transport related to mixing of the sediment through bioturbation (Slomp et al., 1997). At sites with little or no bioturbation, as is the case at our study site (macrofauna were absent, see above), the impact of Mn(II) sorption will be very small. We added this in the model description in the supplements.

Revised text (lines 43 – 44): "Interactions between Mn oxide and ammonium ($NH_4^+$) have also been proposed (Hulth et al., 1999; Thamdrup and Dalsgaard, 2000). The occurrence of this process in marine environments is still debated, however."

Revised text (lines 46 – 47): "… and can adsorb onto Mn oxide minerals (van der Zee et al., 2001)"

Revised text (lines 64 – 65): "… nitrite ($NO_2^-$) and OM is also possible (Kostka et al., 1995; Oldham et al., 2015, 2019; Karolewski et al., 2021) …"

Revised text (line 129 - 130): "… and, in September, for the analysis of $NO_2^-$ and nitrate ($NO_3^-$) were stored at -20°C."

Revised text (lines 144 – 146): "Concentrations of $NO_2^-$ and $NO_3^-$ were measured with a Gallery™ Automated Chemistry Analyzer type 861 (Thermo Fisher Scientific; detection limit of 1 $\mu$mol L$^{-1}$). "

Revised text (line 270 – 271): "In September, $NO_2^-$ and $NO_3^-$ did not exceed the detection limit of 1 µmol L$^{-1}$ and showed no trend with depth."

Revised text (Supplements lines 36 – 38): "Reduction of $MnO_2$ with $NH_4^+$ as discussed by (Hulth et al., 1999) and (Thamdrup and Dalsgaard, 2000) is not incorporated in the model, because the quantitative importance of this reaction is not well known."

Revised text (Supplements lines 39 - 41): "Reduction of dMn(III)-L by $NO_2^-$ as described by Karolewski et al. (2021) is not incorporated in the model. Production of $NO_2^-$ in the anoxic sediment is unlikely and concentrations in September are below the detection limit."

Revised text (Supplements lines 43 – 45): "The dominant effect of adsorption of dissolved Mn(II) to solid phase Mn is related to transport through bioturbation (Slomp et al., 1997). At sites with little or no bioturbation, as is the case here, the effect of Mn(II) adsorption on modelled pore water profiles will be limited."

**Bibliography**

Berg, P., Rysgaard, S. and Thamdrup, B.: "Dynamic modeling of early diagenesis and nutrient cycling. A case study in an artic marine sediment." Am. J. Sci. 303.10: 905-955, 2003.

Canfield, D. E., Thamdrup, B., and Hansen, J. W.: The anaerobic degradation of organic matter in Danish coastal sediments: Iron reduction, manganese reduction, and sulfate reduction, Geochim. Cosmochim. Acta, 57, 3867–3883, https://doi.org/10.1016/0016-7037(93)90340-3, 1993.

Egger, M., Lenstra, W., Jong, D., Meysman, F. J. R., Sapart, C. J., Van Der Veen, C., Röckmann, T., Gonzalez, S., & Slomp, C. P.: Rapid sediment accumulation results in high methane effluxes from coastal sediments. PLoS ONE, 11(8), 1–22. https://doi.org/10.1371/journal.pone.0161609 , 2016

Hagens, M., Slomp, C. P., Meysman, F. J. R. , Seitaj, D. , Harlay, J. , Borges, a V., and Middelburg, J. J.: Biogeochemical processes and buffering capacity concurrently affect acidification in a seasonally hypoxic coastal marine basin. Biogeosciences 12: 1561–1583, 2015.

Hulth, S., Aller, R. C., and Gilbert, F.: Coupled anoxic nitrification/manganese reduction in marine sediments, Geochim. Cosmochim. Acta, 63, 49–66, https://doi.org/10.1016/S0016-7037(98)00285-3, 1999.

Karolewski, J. S., Sutherland, K. M., Hansel, C. M., and Wankel, S. D.: An isotopic study of abiotic nitrite oxidation by ligand-bound manganese (III), Geochim. Cosmochim. Acta, https://doi.org/10.1016/j.gca.2020.11.004, 2020.

Kim, B., Lingappa, U. F., Magyar, J., Monteverde, D., Valentine, J. S., Cho, J., and Fischer, W.: Challenges of Measuring Soluble Mn(III) Species in Natural Samples, Molecules, 27, 1661, https://doi.org/10.3390/molecules27051661, 2022.

Luther, G. W., Sundby, B., Lewis, B. L., Brendel, P. J., and Silverberg, N.: Interactions of manganese with the nitrogen cycle: alternative pathways to dinitrogen, Geochim. Cosmochim. Acta, 61, 4043–4052, 1997.

Luther III, G. W., Karolewski, J. S., Sutherland, K. M., Hansel, C. M., and Wankel, S. D.: The Abiotic Nitrite Oxidation by Ligand-Bound Manganese (III): The Chemical Mechanism, Aquat. Geochem., 27, 207–220, https://doi.org/10.1007/s10498-021-09396-0, 2021.

Madison, A. S., Tebo, B. M., & Luther, G. W.: Simultaneous determination of soluble manganese(III), manganese(II) and total manganese in natural (pore)waters. Talanta, 84(2), 374–381. https://doi.org/10.1016/j.talanta.2011.01.025, 2011

Madison, A. S., Tebo, B. M., Mucci, A., Sundby, B., & Luther III, G. W.: Abundant Porewater Mn(III) Is a Major Component of the Sedimentary Redox System. Science, 341(August), 875–878. https://doi.org/10.5040/9780755621101.0007, 2013

Oldham, V. E., Owings, S. M., Jones, M. R., Tebo, B. M., & Luther, G. W.: Evidence for the presence of strong Mn(III)-binding ligands in the water column of the Chesapeake Bay. Mar. Chem., 171, 58–66. https://doi.org/10.1016/j.marchem.2015.02.008, 2015

Reed, D.C., Gustafsson, B.G. and Slomp, C.P.: "Shelf-to-basin iron shuttling enhances vivianite formation in deep Baltic Sea sediments." Earth and Planet. Sci. Lett. 434: 241-251, 2016.

Richard, D., Sundby, B., and Mucci, A.: Kinetics of manganese adsorption, desorption, and oxidation in coastal marine sediments, Limnol. Oceanogr., 58, 987–996, https://doi.org/10.4319/lo.2013.58.3.0987, 2013.

Shaw, T. J., Sholkovitz, E. R., and Klinkhammer, G.: Redox dynamics in the Chesapeake Bay: The effect on sediment/water uranium exchange, Geochim. Cosmochim. Acta, 58, 2985–2995, https://doi.org/10.1016/0016-7037(94)90173-2, 1994.

Slomp, C. P., Malschaert, J. F. P., Lohse, L., & Van Raaphorst, W.: Iron and manganese cycling in different sedimentary environments on the North Sea continental margin. Science, 17(9), 1083–1117, 1997.

Thamdrup, B. and Dalsgaard, T.: The fate of ammonium in anoxic manganese oxide-rich marine sediment, Geochim. Cosmochim. Acta, 64, 4157–4164, https://doi.org/10.1016/S0016-7037(00)00496-8, 2000.

Thibault de Chanvalon, A. and Luther, G. W.: Mn speciation at nanomolar concentrations with a porphyrin competitive ligand and UV–vis measurements, Talanta, 200, 15–21, https://doi.org/10.1016/j.talanta.2019.02.069, 2019.

Thibault De Chanvalon, A., Luther, G. W., Estes, E. R., Necker, J., Tebo, B. M., Su, J., and Cai, W.-J.: Influence of manganese cycling on alkalinity in the redox stratified water column of Chesapeake Bay, Biogeosciences, 20, 3053–3071, https://doi.org/10.5194/bg-20-3053-2023, 2023.

van Cappellen, Philippe, and Wang, Y.: "Metal cycling in surface sediments: modeling the interplay of transport and reaction." Metal contaminated aquatic sediments. Routledge, 21-64, 2018.
van der Zee, C., van Raaphorst, W., and Epping, E.: Absorbed $Mn^{2+}$ and Mn redox cycling in Iberian continental margin sediments (northeast Atlantic Ocean), J. Mar. Res., 59, 133–166, https://doi.org/10.1357/002224001321237407, 2001.

Żygadłowska, O. M., Venetz, J., Klomp, R., Lenstra, W. K., Van Helmond, N. A. G. M., Röckmann, T., Wallenius, A. J., Martins, P. D., Veraart, A. J., Jetten, M. S. M., and Slomp, C. P.: Pathways of methane removal in the sediment and water column of a seasonally anoxic eutrophic marine basin, Front. Mar. Sci., 10, 1085728, https://doi.org/10.3389/fmars.2023.1085728, 2023.

Żygadlowska, O. M., van Helmond, N.A.G.M., Lenstra, W.K., Klomp, R., Accou, R., Puyk, R., Dickson, A.J., Jetten, M.S.M., and Slomp, C.P.: Seasonal euxinia in a coastal basin: Impact on sedimentary molybdenum enrichments and isotope signatures, Chem. Geol., https://doi.org/10.1016/j.chemgeo.2024.122430, 2024a

Żygadlowska, O. M., Venetz, J., Lenstra, W. K., van Helmond, N.A.G.M., Klomp, R., Röckmann, T., Veraart, A. J., Jetten, M. S. M., and Slomp, C. P.: Ebullition drives high methane emissions from a eutrophic coastal basin, Geochim. Cosmochim. Acta, https://doi.org/10.1016/j.gca.2024.08.028, 2024b.

**Additional changes made**

Where necessary, the citations are changed to comply with the submission guide lines:

In-text citations are re-shuffled to get the right order of publication years in on the following lines: Line 34, line 412 and line 419

The reference list at the end of the manuscript is re-written in accordance with the guidelines.

The following pre-print:
Żygadłowska, O. M., Venetz, J., Lenstra, W. K., van Helmond, N. A. G. M., Klomp, R., Röckmann, T., Veraart, A. J., Jetten, M. S. M., & Slomp, C. P.: High methane emissions from a eutrophic marine coastal basin driven by bubble release from the sediment. EarthArXiv, 2023b

Is published as:
Żygadłowska, O. M., Venetz, J., Lenstra, W. K., Van Helmond, N. A. G. M., Klomp, R., Röckmann, T., & Slomp, C. P.: Ebullition drives high methane emissions from a eutrophic coastal basin. Geochim. Cosmochim. Ac., 384, 1–13. https://doi.org/10.1016/j.gca.2024.08.028, 2024b.

Therefore, the reference Żygadłowska et al., 2023b is substituted for Żygadłowska et al., 2024b. As a result, in text citation Żygadłowska et al., 2023a is changed to Żygadłowska et al., 2023. These changes are made in the following lines:
line 90 – 91, line 96, line 97, line 115, line 141 and line 442

The Fe extraction method followed is correctly described by Kraal et al. (2017) instead of Kraal et al. (2012), therefore this citation is changed in the text in lines 178 – 179 and in the reference list.

Kraal, P., Dijkstra, N., Behrends, T., & Slomp, C. P.: Phosphorus burial in sediments of the sulfidic deep Black Sea: Key roles for adsorption by calcium carbonate and apatite authigenesis. Geochim. Cosmochim. Ac., 204, 140-158, 2017.

Small textual corrections to correct typo's or improve text flow are implemented in the following lines:

Line 105: "one when the water column was mixed ... one when the water column was stratified ..." ; For extra clarification.

Line 125: "... cm ..." ; Typo correction.

Line 127: "... $NH_4^+$ ..." ; $NH_4^+$ was already introduced in the text, so no longer needs brackets.

Line 131: "... (TD Fe and TD Mn) ..." ; Abbreviations were not yet introduced.

Line 179: "The extraction procedure ..." ; For extra clarification.

Line 207: "... including the dynamics of both dissolved ..." ; For extra clarification

Line 213 – 214: "... and were, where relevant, adjusted for environmental characteristics at..." ; Typo corrections.

Line 293: "Porosity values vary ..." ; Typo correction.

Line 394: "... co-occurred ..." ; Typo correction

Lines 406 – 407: "... were completely reduced and dissolved Fe precipitated ..."

Line 418: "... which can explain ..." ; Change of text to improve flow of text.

In supplements equation 3, parameter x is replaced by z, because this is used for depth in the other equations.

Supplements line 62: "… $\mu mol\ L^{-1}$…" ; adjusted to be in line with the style used in the rest of the manuscript.

Supplements line 67: " … until 2016 (4 years before …" & supplements line 70: "For the last 4 years…" ; The timing of the change in sedimentation rate induced in the model was 4 years before the end of the model (hence 2016), instead of the 6 years (2014) reported earlier.

Supplements line 38: "Reduction of dMn(III)-L …"; Adjusted to be in line with the abbreviations in the rest of the text.

Figure S4 is replaced by a figure where the panels are marked with lower case instead of capital letters.

Several typo's in the supplementary tables are corrected.

References to the supplementary figures are updated in the supplementary figure captions and throughout the main and supplementary texts.